# Inhibiting stromal Class I HDACs curbs pancreatic cancer progression

Gaoyang Liang [1], Tae Gyu Oh[1,10], Nasun Hah[2], Hervé Tiriac [3], Yu Shi[4,11], Morgan L. Truitt[1], Corina E. Antal [1,12], Annette R. Atkins[1], Yuwenbin Li[1], Cory Fraser[5], Serina Ng[6], Antonio F. M. Pinto[7], Dylan C. Nelson[1], Gabriela Estepa[1], Senada Bashi[1], Ester Banayo[1], Yang Dai[1], Christopher Liddle[8], Ruth T. Yu [1], Tony Hunter [4], Dannielle D. Engle[9], Haiyong Han [6], Daniel D. Von Hoff[5,6], Michael Downes[1] ✉ & Ronald M. Evans [1] ✉

Oncogenic lesions in pancreatic ductal adenocarcinoma (PDAC) hijack the epigenetic machinery in stromal components to establish a desmoplastic and therapeutic resistant tumor microenvironment (TME). Here we identify Class I histone deacetylases (HDACs) as key epigenetic factors facilitating the induction of pro-desmoplastic and pro-tumorigenic transcriptional programs in pancreatic stromal fibroblasts. Mechanistically, HDAC-mediated changes in chromatin architecture enable the activation of pro-desmoplastic programs directed by serum response factor (SRF) and forkhead box M1 (FOXM1). HDACs also coordinate fibroblast pro-inflammatory programs inducing leukemia inhibitory factor (LIF) expression, supporting paracrine pro-tumorigenic crosstalk. HDAC depletion in cancer-associated fibroblasts (CAFs) and treatment with the HDAC inhibitor entinostat (Ent) in PDAC mouse models reduce stromal activation and curb tumor progression. Notably, HDAC inhibition (HDACi) enriches a lipogenic fibroblast subpopulation, a potential precursor for myofibroblasts in the PDAC stroma. Overall, our study reveals the stromal targeting potential of HDACi, highlighting the utility of this epigenetic modulating approach in PDAC therapeutics.

With a dismal survival rate, PDAC is predicted to become the second most lethal cancer by 2030[1,2]. The poor prognosis and therapeutic resistance have been in part attributed to the prominent activated stroma resulting from a desmoplastic response induced by the transformed pancreatic epithelium[3,4]. The desmoplastic response is largely due to the activation of fibroblast-like cells, including pancreatic stellate cells (PSCs), the major resident fibroblasts in pancreatic stroma. Underlying PSC activation is a switch of quiescence-associated transcriptional programs (e.g. a lipogenic program) to pro-desmoplastic ones including those driving

[1]Gene Expression Laboratory, Salk Institute for Biological Studies, La Jolla, CA 92037, USA. [2]Next Generation Sequencing Core, Salk Institute for Biological Studies, La Jolla, CA 92037, USA. [3]Department of Surgery, University of California San Diego, La Jolla, CA 92093, USA. [4]Molecular and Cell Biology Laboratory, Salk Institute for Biological Studies, La Jolla, CA 92037, USA. [5]HonorHealth Scottsdale Osborn Medical Center and Shea Medical Center, Scottsdale, AZ 85260, USA. [6]Molecular Medicine Division, The Translational Genomic Research Institute, Phoenix, AZ 85004, USA. [7]Mass Spectrometry Core, Salk Institute for Biological Studies, La Jolla, CA 92037, USA. [8] Storr Liver Centre, Westmead Institute for Medical Research and Sydney Medical School, University of Sydney, Westmead Hospital, Westmead, NSW 2145, Australia. [9]Regulatory Biology Laboratory, Salk Institute for Biological Studies, La Jolla, CA 92037, USA. [10]Present address: Department of Oncology Science, OU Health Stephenson Cancer Center, University of Oklahoma Health Sciences Center, Oklahoma City, OK 73117, USA. [11]Present address: Bristol Myer Squibb, 10300 Campus Point Drive, Suite 100, San Diego, CA 92121, USA. [12]Present address: Department of Pharmacology, University of California San Diego, La Jolla, CA 92093, USA. ✉e-mail: downes@salk.edu; evans@salk.edu

myofibroblast transdifferentiation and proliferation[3,5-7]. The accumulation of activated PSCs or CAFs, as well as CAF-secreted extracellular matrix (ECM) molecules and paracrine growth factors, establishes a tumor-promoting[8-11], immunosuppressive[12-14] and drug-resistant TME[15,16]. The association of activated stroma with poor patient prognosis[17] further suggests stromal targeting strategies as a potential adjuvant approach to improve the therapeutic outcome in PDAC therapy[7,18]. Paradoxically, the PDAC stroma also has a tumor-restraining effect, as near-complete stromal fibroblast depletion results in more aggressive tumors[19-21]. Recent findings that stromal fibroblasts from PDAC consist of heterogeneous transcriptional and phenotypic subpopulations, including myofibroblastic (also called myCAF), inflammatory (iCAF) or adventitial, and other subpopulations, implicate potential subpopulation-specific functions in the fibroblasts[13,22-25]. Approaches designed to reduce pro-desmoplastic and pro-tumorigenic features in stromal fibroblasts without ablating the tumor-stroma architecture could provide a potentially safe and effective strategy to improve therapeutic outcome[7,18]. Supporting this notion, we previously showed that expression of vitamin D receptor (VDR) in PSCs allows vitamin D analogs to reprogram activated PSCs to enhance chemotherapy efficacy[6].

The ability of VDR-mediated stromal modulation to potentiate PDAC therapy raised the question of whether additional epigenetics-based stromal targeting approaches have therapeutic potential. HDAC inhibitors are a class of epigenetic modulators currently being explored as cancer therapies[26,27]. While HDACs in pancreatic tumor cells have been associated with PDAC development[28-34], the role of HDACs in stromal fibroblasts is poorly understood and the therapeutic potential of HDACi, in particular, the stromal modulating potential remains to be fully explored. Here we identify that Class I HDACs, a set of chromatin modifiers that coordinate transcriptional regulation, are essential for activating the pro-desmoplastic and pro-tumorigenic transcriptional programs that drive stromal activation and PDAC progression. The effects of HDACi on reducing stromal activation and pro-tumorigenicity highlight the potential of HDACi as a stromal targeting strategy in PDAC therapy.

## Results

### HDACs facilitate PSC activation by inducing SRF/FOXM1-directed pro-desmoplastic transcriptional programs

To test the potential of HDACi to regulate stromal activation in PDAC, we initially determined the ability of entinostat (Ent), a Class I HDAC inhibitor under Phase III studies[26,35,36], to modulate PSC activation in vitro (Fig. 1a). In the absence of any drug treatment, culturing freshly isolated PSCs from normal mouse pancreas for 3-6 days induces an activated phenotype, characterized by the loss of quiescence-specific cytoplasmic lipid droplets (Supplementary Fig. 1a, b), the induction of the myofibroblast marker α-SMA (α-smooth muscle actin, gene name Acta2), and active proliferation as shown by induced expression of proliferation marker Ki67 (gene name, Mki67) and the ability to incorporate EdU (Fig. 1b, c). Moreover, multiple myofibroblast and proliferation genes, as parts of pro-desmoplastic transcriptional programs, were dramatically induced during PSC activation (Supplementary Fig. 1c, d). Although HDACs are conventionally deemed as transcriptional repressors for their capacity to remove histone acetylation that activate transcription, treating PSCs with Ent from day 1 blocked the induction of the myofibroblast and proliferation genes during culture-induced activation, including α-SMA and Ki67 (Fig. 1b, c and Supplementary Fig. 1c, d), with individual targets showing differential sensitivity to Ent (Supplementary Fig. 1e). Consistent with reduced expression of proliferation markers, Ent treatment suppressed EdU incorporation (Fig. 1b, c) and reduced PSC cell number during culture-induced activation in a concentration-dependent manner (Supplementary Fig. 1f); while no increase in apoptosis was detected (Supplementary Fig. 1g, h). Knockdowns of Class I HDACs,

most notably HDAC1 and 2, similarly attenuated the induction of myofibroblast and proliferation genes during PSC activation (Supplementary Fig. 1i), further supporting a central role for HDACs in inducing the pro-desmoplastic programs. Interestingly, limiting Ent treatment to the final 3 days of PSC activation largely recapitulated the effects of continuous treatment (Supplementary Fig. 1c, d, V-E compared to E-E treatment), suggesting that Ent can reverse the activated pro-desmoplastic programs; whereas few residual effects were evident in day 6 when Ent treatment was restricted to the first 2 days of treatment (E-V compared to V-V treatment). Collectively, these findings suggest that PSC activation is an epigenetic transition facilitated by Class I HDACs and suppressed by HDACi.

To delineate HDAC-regulated transcriptional programs, we compared the genome-wide expression changes induced by in vitro activation of PSCs in the presence and absence of Ent. While the transcriptional changes seen with PSC activation were extensive, they were broadly inhibited by the presence of Ent, including a comprehensive suppression of the pro-desmoplastic transcriptional programs controlling genes important for proliferation and those encoding ECM, cytoskeleton, adhesion and signaling components typical for myofibroblasts (Fig. 1d, e, Supplementary Fig. 2a). Gene set enrichment analysis (GSEA) confirmed that the gene sets induced by in vitro activation were enriched for those downregulated by Ent treatment (Fig. 1f), with approximately half of the genes induced during in vitro PSC activation being significantly reduced by Ent (Fig. 1g). Gene ontology (GO) analysis revealed that these genes were functionally enriched for categories related to PSC activation including proliferation and myofibroblast identity (Fig. 1h), highlighting a potential for HDACi to disable the functional transcriptional programs driving PSC activation. Furthermore, reciprocally, gene sets repressed by in vitro activation were enriched for those upregulated by Ent treatment (Fig. 1f), with the repression of ~30% of the downregulated genes in PSC activation being reversed by Ent (Fig. 1i). Interestingly, the de-repression of genes related to lipid metabolism, including Fabp4 (fatty acid associated protein 4) (Fig. 1d, e, j), a marker for quiescent PSCs and normal stroma[6,17], suggests the retention of at least part of the quiescence-associated lipogenic programs by Ent despite the visual loss of lipid droplets (Supplementary Fig. 1a, b). Together, these findings identify HDACs as key coordinators switching quiescence-associated transcriptional programs to pro-desmoplastic ones in PSC activation.

The concerted regulation of hundreds of functionally convergent genes under the pro-desmoplastic programs led to the speculation that Ent affects the activity of TFs or coregulators directing these programs. To explore this possibility, we interrogated curated databases to identify TFs or cofactors that could preferentially bind to the subset of genes induced by in vitro activation and antagonized by Ent (Fig. 1g and Supplementary Data 1). Prioritizing this list of transcriptional regulators based on characterized or implicated functions in myofibroblasts[37-39] identified FOXM1, SRF, YAP1 and TEAD family TFs as candidate drivers (Fig. 2a). Encouragingly, the expression of these transcriptional regulators was synchronous with their putative targets, being induced by in vitro activation and suppressed by Ent treatment (Fig. 1e) and HDAC depletion (Supplementary Fig. 2b).

To assess the functions of these putative transcriptional drivers, we determined the consequences of FOXM1 and SRF knockdowns (Fig. 2b) on the pro-desmoplastic transcriptional programs in activated PSCs. Of the genes induced by in vitro activation, ~10% showed reduced expression with the depletion of FOXM1 (Fig. 2c). These FOXM1-dependent genes were enriched in components of cell cycle progression (Fig. 2b-e), in agreement with the established role of FOXM1 in regulating proliferation[40]. Loss of SRF compromised the induction of ~25% of the in vitro activation gene set, including the majority of FOXM1-dependent proliferative genes as well as FOXM1-insensitive myofibroblastic genes (Fig. 2b-d, f, g), consistent with the role of SRF in directing myofibroblast activation[37,38]. Interestingly, the

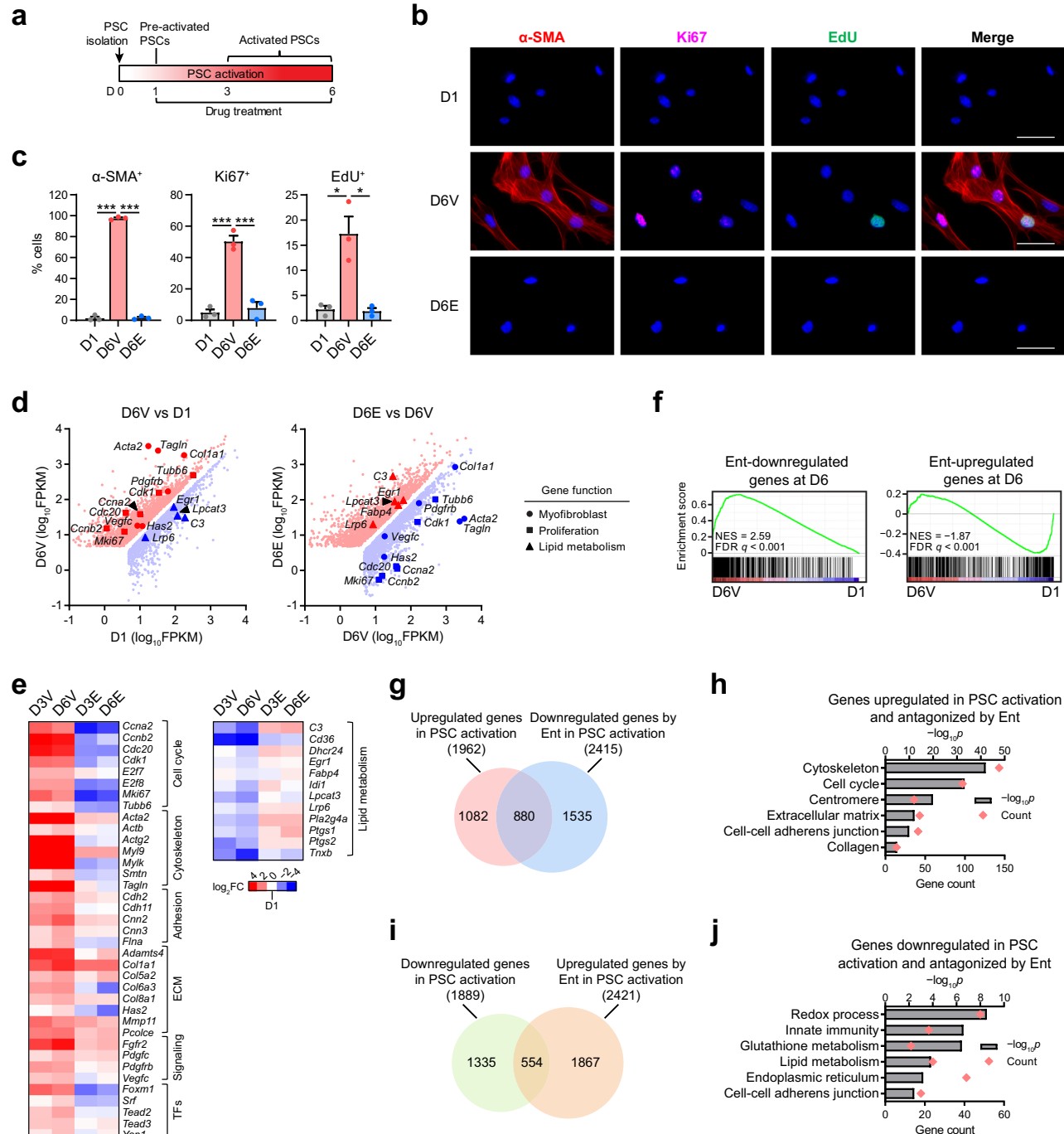

**Fig. 1 | HDACi suppresses PSC activation via transcriptional regulation.**
**a** Scheme of in vitro PSC activation with drug treatment. **b, c** Representative images (**b**) and quantifications (**c**) of immunofluorescence staining for α-SMA, Ki67 and EdU in PSCs at day (D) 1 and 6 under vehicle (Veh, D6V) or Ent treatment (D6E; 5 μM). Scale bar, 50 μm. $n = 3$ independent samples. Data are presented as mean values ± SEM. *$p < 0.05$, = 0.044 (EdU⁺, D6V vs D1), 0.011 (EdU⁺, D6E vs D6V); ***$p < 0.001$ (others). Two-sided $t$-test. **d** Scatter plots from RNA-seq data showing genes significantly upregulated (red) and downregulated (blue) in D6V compared to D1, and those in D6E compared to D6V, with representative genes functionally related to myofibroblast identity (circle), proliferation (square) and lipid metabolism (triangle) highlighted. $n = 3$ independent samples. False discovery rate (FDR)

$q < 0.05$. FPKM, fragments per kilobase per million reads. **e** Heatmap showing the expression fold-change (FC) in PSC samples for selected functional genes in PSC activation with or without Ent. **f** GSEA plots showing the top 500 Ent-downregulated and -upregulated genes at D6 are respectively in genes induced and repressed in PSC activation. NES, normalized enrichment score. **g, h** Venn diagram comparing genes upregulated in PSC activation and those downregulated by Ent at D6 (**g**) and selected ontology terms enriched in the overlap genes with $p$-values and gene counts from GO analysis (**h**). **i, j** Venn diagram comparing genes downregulated in PSC activation and those upregulated by Ent at D6 (**i**) and selected ontology terms enriched in the overlap genes from GO analysis (**j**). GO analysis (**h, j**), one-sided Fisher's exact test. Source data are provided in a Source Data file.

induction of *Foxm1* was also dependent on SRF (Fig. 2b, d), suggesting a hierarchical TF axis coordinating to regulate the pro-desmoplastic programs. Moreover, the expression of *Hdac1, 2* and *3* appears to depend on FOXM1 and/or SRF, indicating potential feed-back regulation between Class I HDACs and the TFs (Supplementary Fig. 2c). Overall, our findings suggest that HDACs coordinate the induction of the pro-desmoplastic transcriptional programs in part through the activities of SRF and FOXM1.

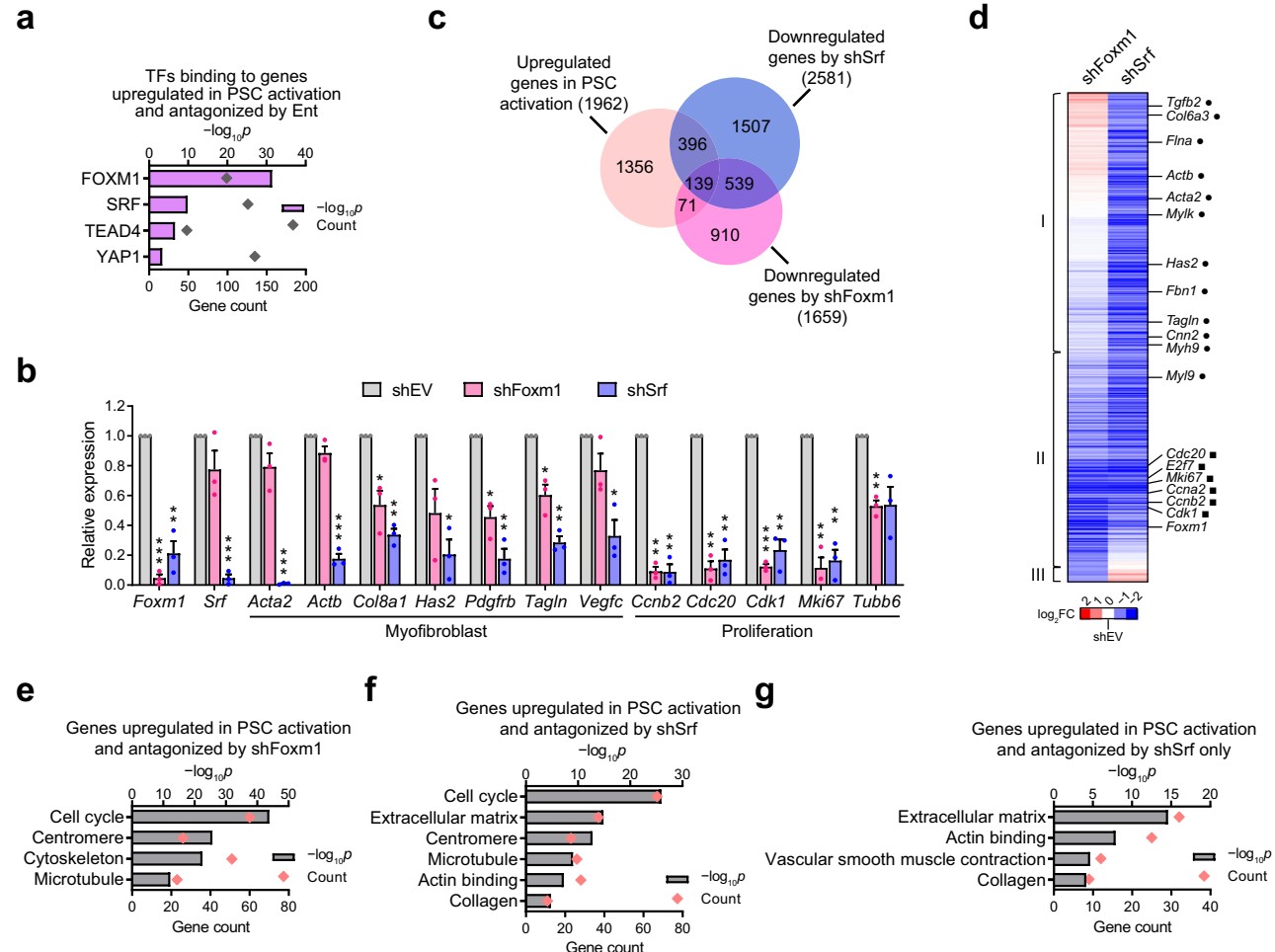

**Fig. 2 | The SRF-FOXM1 TF axis mediates HDAC-coordinated transcriptional programs in PSC activation. a** Enrichment of TF binding sites at genes induced in PSC activation and suppressed by Ent. One-sided Fisher's exact test. **b** RT-qPCR results showing the expression of *Foxm1*, *Srf* and selected functional markers upon shRNA-mediated inhibition (sh-), compared to empty vector (shEV). $n = 3$ independent samples. Data are presented as mean values ± SEM. *$p < 0.05$; **$p < 0.01$; ***$p < 0.001$. shFoxm1 vs shEV, $p < 0.001$ (*Foxm1*, *Cdk1*), = 0.040 (*Col8a1*), 0.018 (*Pdgfrb*), 0.030 (*Tagln*), 0.001 (*Ccnb2*), 0.003 (*Cdc20*), 0.006 (*Mki67*), 0.007 (*Tubb6*); shSrf vs shEV, $p = 0.011$ (*Foxm1*), <0.001 (*Srf*, *Acta2*), = 0.002 (*Actb*), 0.004 (*Col8a1*), 0.015 (*Has2*), 0.007 (*Pdgfrb*, *Cdc20*, *Mki67*), 0.003 (*Tagln*, *Ccnb2*), 0.024 (*Vegfc*), 0.009 (*Cdk1*). Two-sided *t*-test. **c** Venn diagram showing the distribution of

genes significantly upregulated in activated PSCs and those downregulated by shFoxm1 or shSrf (FC > 2) as identified by RNA-seq. $n = 2$ independent samples. **d** Heatmap from hierarchical clustering showing the effects of FOXM1 and SRF depletion on genes induced by in vitro activation and suppressed by either TF depletion, including markers for myofibroblast (dot) and proliferation (square). Genes suppressed by shSrf only, by both shSrf and shFoxm1, and by shFoxm1 only (III) are respectively grouped in I, II, and III. **e–g** Selected ontology terms enriched in genes induced in PSC activation and suppressed by shFoxm1 (**e**), by shSrf (**f**), and by shSrf only but not shFoxm1 (**g**). TF enrichment (**a**) and GO analyses (**e–g**), one-sided Fisher's exact test. Source data are provided in a Source Data file.

## HDACs coordinate chromatin changes in PSC activation

To delve into the mechanism of how HDACs coordinate the transcriptional programs in PSC activation, the genome-wide changes in chromatin accessibility were mapped with assay for transposase-accessible chromatin using sequencing (ATAC-seq)[41]. In vitro activation led to an ~80% increase in accessible sites, including increases in genic and intergenic (~90%), as well as promoter sites (~15%) (Fig. 3a, D3V compared to D1, Supplementary Fig. 3a). Notably, these changes were primarily due to the addition of de novo sites (93%), with only a marginal loss (14%) of pre-existing sites (Fig. 3b). In total, accessibility at over 55,000 sites was increased (> 2 folds), with an average 4-fold gain in accessibility (Fig. 3c–f). Together, these findings associate PSC activation with increased chromatin accessibility across the genome.

To functionally connect chromatin changes with transcriptional outcomes, accessible sites were associated with the most adjacent genes. Notably, >60% of genes transcriptionally upregulated by in vitro activation harbored one or more genomic sites with significantly increased accessibility (> 4 folds; Supplementary Fig. 3b, c). GSEA

revealed that genes associated with these accessible sites, as well as each subset specifically linked to the genic, intergenic or promoter sites, were enriched for those upregulated during PSC activation (Fig. 3g, Supplementary Fig. 3d). Genomic sites with increased accessibility in promoters correlated with greater transcriptional induction (Supplementary Fig. 3e), potentially reflecting promoter function and/or better prediction of the nearest gene method in associating promoters with their regulating targets. Meanwhile, investigation of the genomic sites at or near genes induced by PSC activation and suppressed by Ent also revealed pervasive increased accessibility post-activation (Supplementary Fig. 4a), including subsets of genes related to myofibroblast identity and proliferation, as well as the relevant TF genes (Fig. 3d, h–j, Supplementary Fig. 4b, c, Supplementary Data 2). The presence of de novo sites in putative regulatory regions such as promoters (Fig. 3h) implicates the prerequisite of open chromatin for induced transcription during PSC activation. Motif analysis with curated TF binding sites revealed an enrichment of SRF and FOXM1 motifs in sites with increased accessibility post-activation (Supplementary

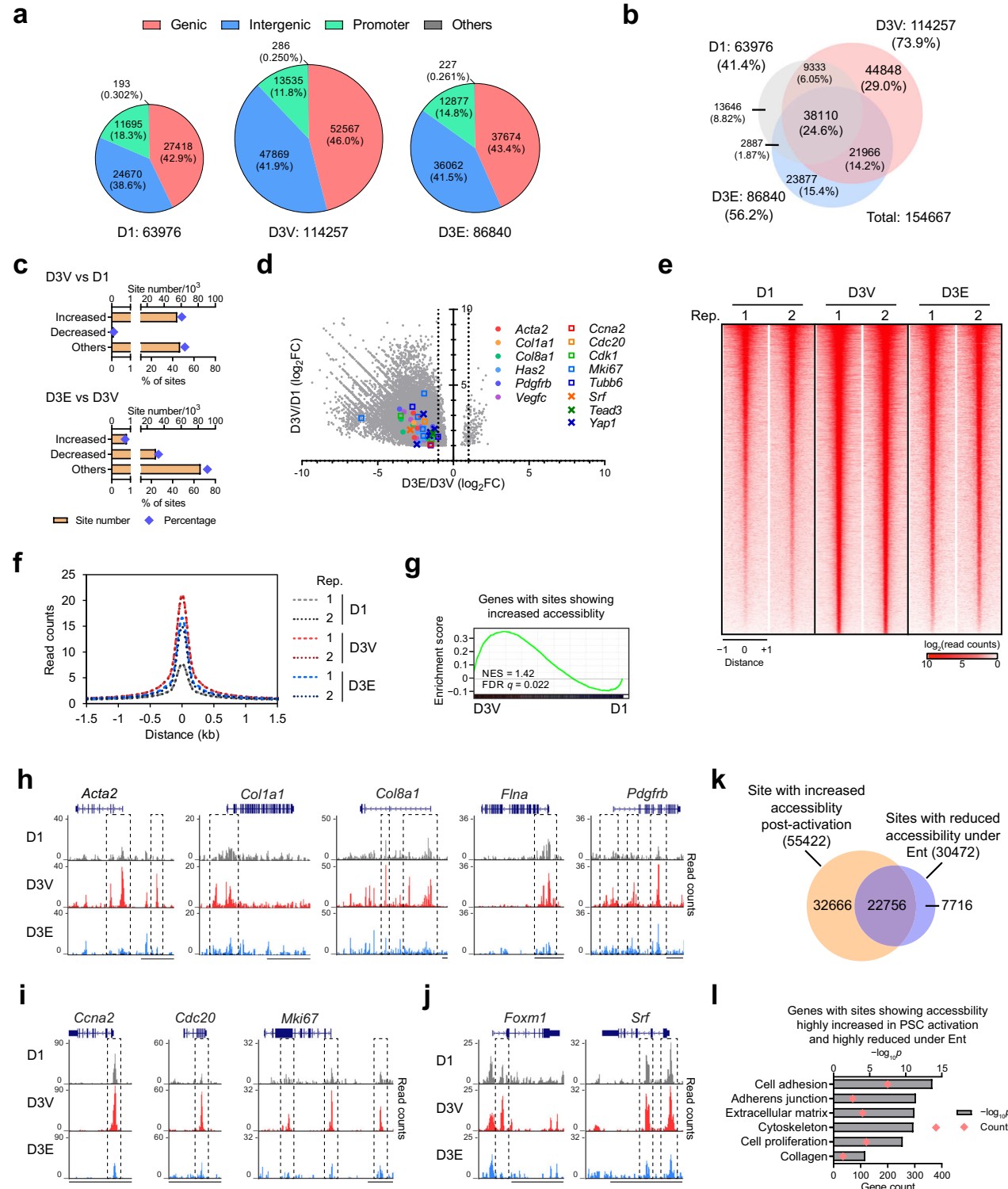

Fig. 4d). Genes with induced accessible sites containing these TF motifs are enriched in gene sets involved in myofibroblast and proliferation functions (Supplementary Fig. 4e), consistent with the roles of these TFs in directing the pro-desmoplastic transcriptional programs (Fig. 2). In addition, the top de novo motifs identified at sites with increased accessibility upon activation were best matched to the AP-1, TEAD and RUNX motifs (Supplementary Fig. 4f). The roles of these TF families in PSC activation remain to be investigated.

The above findings associate the induced transcriptional programs with increased chromatin accessibility in activated PSCs. Given that Ent

was able to suppress the transcriptional changes, we explored whether this ability was driven by changes in chromatin accessibility. Inclusion of the HDAC inhibitor during in vitro activation led to a ~25% reduction in accessible chromatin sites compared to activated PSCs (Fig. 3a, b). Around 30,000 sites in activated PSCs, including ~23,000 sites with increased accessibility post-activation, displayed reduced accessibility under Ent (Fig. 3c, i). Importantly, activation-induced increases in accessibility were reduced at the majority of sites (>93%) with an average ~50% reduction in accessibility (Fig. 3d–f). Moreover, genomic sites with accessibility highly increased post-activation and highly

**Fig. 3 | Ent treatment restricts chromatin opening during PSC activation.**
**a** Numbers and percentages of accessible sites detected by ATAC-seq and their genomic annotations in pre-activated PSCs (D1) and PSCs at D3 under Veh (D3V) or Ent treatment (D3E, 5 μM). *n* = 2 independent samples. **b** Venn diagrams showing the distributions of accessible sites among PSC samples with the percentages relative to the total detected sites. **c** Numbers of sites with significantly increased or decreased accessibility (FC > 2, FDR *q* < 0.05) and those without significant changes (others) post-activation (D3V vs D1) and under Ent treatment (D3E vs D3V), along with the percentages relative to D3V. **d** Scatter plot showing the accessibility changes post-activation (*y*-axis) and under Ent treatment (*x*-axis) at genomic sites (22,879) with accessibility significantly increased in PSC activation (FC > 2, FDR *q* < 0.05) and changed by Ent (FDR *q* < 0.05), including sites at selected PSC activation markers and TFs. Dotted lines, FC of 2 (D3E vs D3V). **e, f** Heatmap showing the normalized ATAC-seq read counts (**e**) and histogram showing average

normalized read counts (**f**) for genomic sites (55,422) showing significantly increased accessibility post-activation in individual replicates (Rep.) of PSC samples. **g** GSEA plots showing genes upregulated post-activation are enriched in genes with genomic sites showing highly increased accessibility. **h–j** Genome browser tracks for selected myofibroblast (**h**), proliferation (**i**) and TF (**j**) gene loci in representative PSC samples with genomic sites displaying differential accessibility highlighted (box). Scale bar, 10 kb. **k** Venn diagram showing the distribution of genomic sites with significantly increased accessibility post-activation (FC > 2, FDR *q* < 0.05) and those with significantly reduced accessibility under Ent (FC > 2, FDR *q* < 0.05). **l** Representative enriched ontology terms for genes with sites showing accessibility highly increased post-activation (FC > 4, FDR *q* < 0.05) and highly reduced by Ent (FC > 4, FDR *q* < 0.05). GO analysis, one-sided Fisher's exact test. Source data are provided as a Source Data file.

reduced by Ent were associated with genes functioning in myofibroblast identity and proliferation (Fig. 2l). In particular, reduced accessibility by Ent was seen throughout the gene set showing reduced induction in PSC activation under Ent treatment (Supplementary Fig. 4a), including the myofibroblast, proliferation, and the relevant TF genes (Fig. 2d, h–j, Supplementary Fig. 4b, c, Supplementary Data 2). In addition, no consistent correlation between chromatin accessibility and transcriptional outputs was found at the lipid metabolism-related genes downregulated by in vitro activation and upregulated by Ent (Supplementary Fig. 4g, h, Supplementary Data 2), indicating distinct mechanisms in regulating these genes. Together, these findings demonstrate the ability of HDACi to interrupt the chromatin changes required for the activation of pro-desmoplastic transcriptional programs, supporting a role of HDACs in coordinating chromatin establishment in PSC activation.

## HDACi suppresses CAF activation and TGF-β- and TNF-α-induced responses
As CAFs from PDAC usually carry phenotypic features like in activated PSCs[3,7], we explored whether HDACi could similarly regulate the transcriptional programs in CAFs derived from a mouse PDAC model and human PDAC patients. Ent treatment downregulated a similar set of myofibroblast and proliferation markers, as well as the master TFs, SRF and FOXM1, in mouse and human CAFs (Fig. 4a–c, Supplementary Fig. 5a). Reduced proliferation was seen under Ent treatment in CAFs in a dose-dependent manner (Supplementary Fig. 5b), while no induction of apoptosis was observed (Supplementary Fig. 5c). Meanwhile, Ent also upregulated a similar set of lipid-related genes (e.g., *Fabp4/FABP4*) in mouse and human CAFs (Fig. 4a, b), largely replicating the murine PSC model (Fig. 1d, e). Genes downregulated by Ent in CAFs showed functional enrichment for proliferation and biological processes related to myofibroblast function and identity, while those upregulated by Ent were enriched in lipid metabolism and other biological processes (Fig. 4d), further confirming the ability of HDACi to reverse the switch of transcriptional programs leading to CAF activation. Furthermore, Ent retained the capacity to downregulate the pro-desmoplastic programs in CAFs cultured in the 3D condition, which is known to reduce myofibroblastic phenotypes in PSCs/CAFs[23], indicating that the HDACi-directed effect on the pro-desmoplastic programs is largely independent of 2D or 3D culture condition (Supplementary Fig. 5d). Interestingly, concurrent with the downregulation of myofibroblast and proliferation genes in 3D culture, reduction of *Hdac1* and *2* expression was also seen, consistent with the importance of HDAC activities in promoting the pro-desmoplastic programs (Supplementary Fig. 5e). In addition, in 3D culture, Ent persistently upregulated the set of lipid-related genes as in the 2D condition (Supplementary Fig. 5f), again suggesting the transcriptional effects from HDACi on CAF phenotypes are largely independent of those from 3D culture.

Furthermore, Ent systematically downregulated genes involved in TGF-β signaling (Fig. 4d, e), a pro-desmoplastic pathway inducing

myofibroblastic transdifferentiation (Supplementary Fig. 5g)[13,22,42], indicating a key role of HDACs in integrating environmental signal in stromal fibroblasts. To interrogate the role of HDACs in regulating the TGF-β pathway, we investigated how Ent affects the TGF-β response at the transcription level in patient-derived CAFs. In general, Ent treatment attenuated the effects of TGF-β, antagonizing both transcriptional activation (median fold change reduced from 1.6 to 1.1) and repression (median fold change raised from 0.36 to 0.10) (Fig. 3f). Notably, Ent downregulated ~60% of TGF-β induced targets with the effects largely epistatic to TGF-β (Fig. 3g). This set of genes were functionally enriched for terms related to myofibroblast identity, and included key markers such as α-SMA (*ACTA2*) and transgelin (gene name *TAGLN*) (Fig. 3g–i). In addition, Ent treatment also blocked the TGF-β-directed induction of myofibroblast genes in mouse CAFs and PSCs (Supplementary Fig. 5h, i), again demonstrating the importance of HDACs in facilitating the TGF-β-induced pro-desmoplastic response in stromal fibroblasts.

Similarly, Ent treatment systematically downregulated genes involved in TNF-α signaling (Fig. 4d, e), a paracrine pathway active in the TME and capable of inducing inflammatory/pro-tumorigenic phenotype in stromal fibroblasts[22,43] (Supplementary Fig. 5j). The TNF-α targets downregulated by Ent include LIF (Fig. 4a–c, Supplementary Fig. 5d), a pro-tumorigenic cytokine that triggers tumor STAT3 pathway and contributes to PDAC progression and therapeutic resistance[9,44], indicating the potential of HDACi to reduce the pro-tumorigenicity of CAF secretome. Moreover, the overall transcriptional changes in response to TNF-α treatment were attenuated by Ent in CAFs. The median TNF-α-mediated transcription activation was reduced from 2.3 to 1.5-fold by Ent, while the TNF-α-mediated repression alleviated from 0.44 to 0.16-fold (Fig. 4j). Interestingly, the effects of Ent were dominant over TNF-α stimulation on >50% of the TNF-α-induced gene set (Fig. 4k). The TNF-α-induced genes suppressed by Ent showed functional enrichment for terms related to pro-inflammatory response, including the NF-κB and other immune pathways, and included major functional effectors of the TNF-α pathway, such as LIF (Fig. 4k–m). The TNF-α-mediated induction of similar gene sets was also blocked by Ent treatment in mouse CAFs and PSCs (Supplementary Fig. 5k, l). Taken together, these data demonstrate that HDACi can not only reverse the switch of fibroblast activation transcriptional programs in CAFs, but also antagonize the pro-desmoplastic and pro-inflammatory responses driven by TME signals, implicating a potential to reduce tumor promoting effects in PDAC stroma.

## HDACi reduces CAF-mediated pro-tumorigenic signaling
The capacity of Ent to repress *LIF* expression prompted an investigation of whether HDACi reduces the tumor promoting activities of CAFs through modulating paracrine signaling. Concurrent with lower *Lif/LIF* expression (Fig. 5a, b), Ent treatment reduced the abundance of LIF by ~75% in the conditioned media (CM) from human and mouse CAF

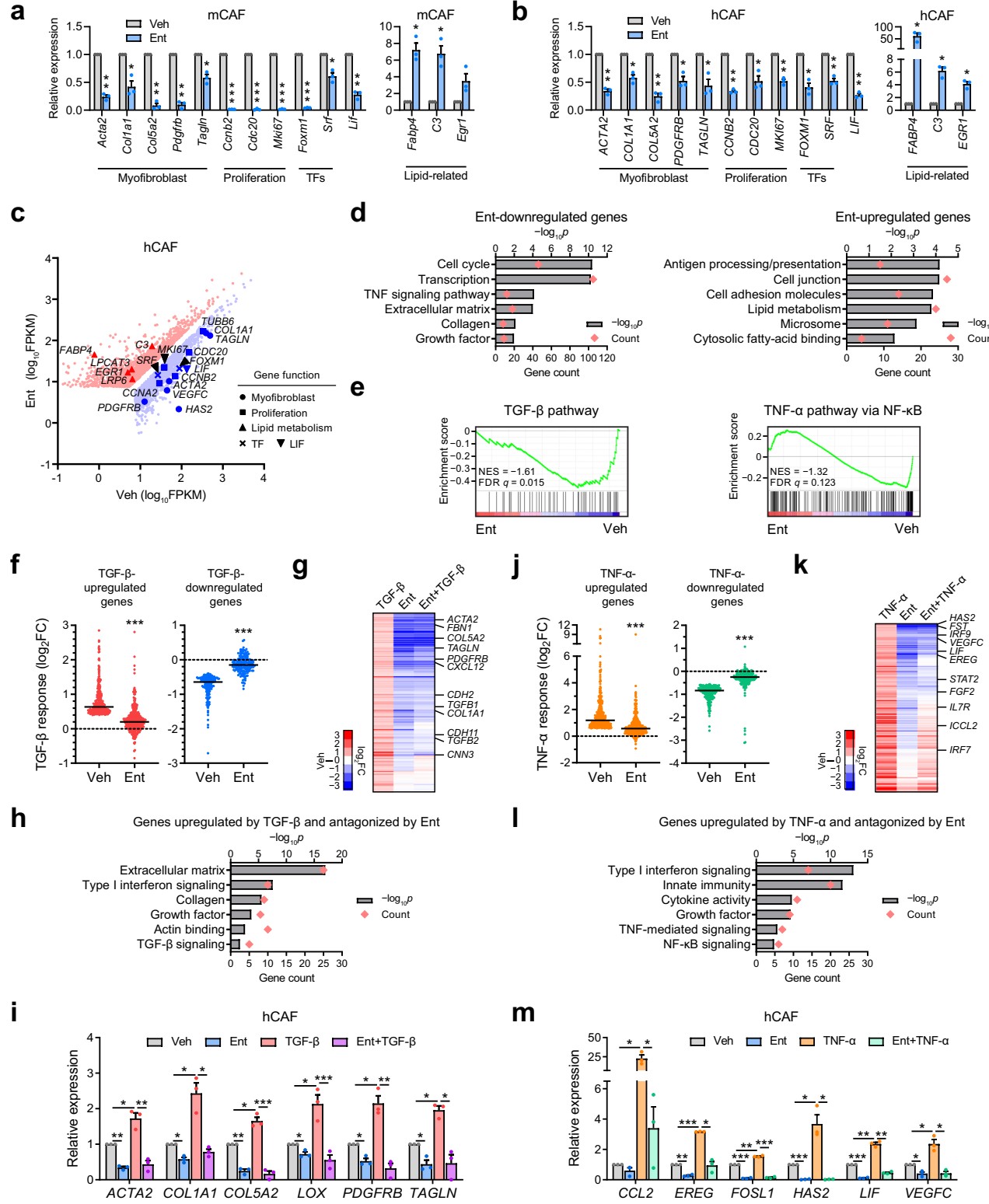

cultures (Fig. 5c, d). The reduction in CAF cell number under Ent treatment was estimated to account for up to 40–60% of the LIF reduction, while the average LIF secretion per cell was estimated to be ~40–60% lower under Ent treatment than in control CAF cultures (Supplementary Fig. 6a, b).

To evaluate the impacts of HDACi on the pro-tumorigenicity of CAF secretomes, conditioned media (CM) was harvested from Veh- and Ent-treated CAFs, processed by centrifugal filtration to deplete the small-molecule fraction (<3 kDa) containing Ent, and analyzed for the capacity to activate tumorigenic STAT3 pathway and to support anchor-independent spheroid formation in PDAC cells. Exposure of PDAC cells to conditioned media (CM) from Ent-treated CAFs led to reduced phosphorylated STAT3 (pSTAT3) (Fig. 5e, f), a marker for active LIFR-STAT3 pathway[9], indicating reduced pro-tumorigenic signaling from CAFs under HDACi. The marked reduction in pSTAT3 by treatment with an inhibitory anti-LIF antibody (α-LIF) implicates LIF as

**Fig. 4 | Ent suppresses CAF activation and TGF-β- and TNF-α-induced responses. a, b** RT-qPCR data showing the expression of representative functional genes after 2 d Ent treatment (10 μM) in mouse (m) (imCAF1, **a**) and human (h) CAF (ONO, **b**) cells. **c** Scatter plot from RNA-seq data showing genes significantly upregulated (red, 2080) or downregulated (blue, 2502) by Ent (10 μM, 2 d) in ONO with functional genes highlighted. FDR *q* < 0.05. **d** Selected ontology terms enriched in the top 500 Ent-downregulated or -upregulated genes. **e** GSEA plots showing the enrichment of TGF-β and TNF-α pathway components in Ent-downregulated genes. **f** Dot plots showing the TGF-β (1 ng/ml, 2 d) induced expression changes under Veh or Ent treatment (10 μM, 2 d) at TGF-β-upregulated (333) or -downregulated (170) genes in ONO. **g** Heatmap from hierarchical clustering showing the expression changes by TGF-β, Ent or both at genes induced by TGF-β and sensitive to Ent-directed suppression. **h** Representative ontology terms enriched in the gene set in

**g. i** RT-qPCR results confirming the Ent effect on selected TGF-β-induced genes in ONO. **j** Dot plots showing the TNF-α (10 ng/ml, 8 h) induced expression changes under Veh or Ent treatment (10 μM, 40 h pre-treatment plus 8 h concurrent treatment with TNF-α) at TNF-α-upregulated (398) or -downregulated (189) genes in hCAFs (YAM). **k** Heatmap showing the expression changes under the treatments of TNF-α, Ent or both at genes upregulated by TNF-α and sensitive to Ent-directed suppression. **l** Representative ontology terms enriched in the gene set in **k. m** RT-qPCR results confirming the Ent effect on selected TNF-α-induced genes in YAM. RT-qPCR (**a**, **b**, **i**, **m**), *n* = 3 independent samples; data are presented as mean values ± SEM. RNA-seq, *n* = 2 (ONO) and 3 (YAM) independent samples. Dot plots (**f**, **j**): bars, medians. *\*p* < 0.05; *\*\*p* < 0.01; *\*\*\*p* < 0.001. Two-sided *t*-test. GO analysis (**d**, **h**, **l**), one-sided Fisher's exact test. Source data including *p*-values are provided as a Source Data file.

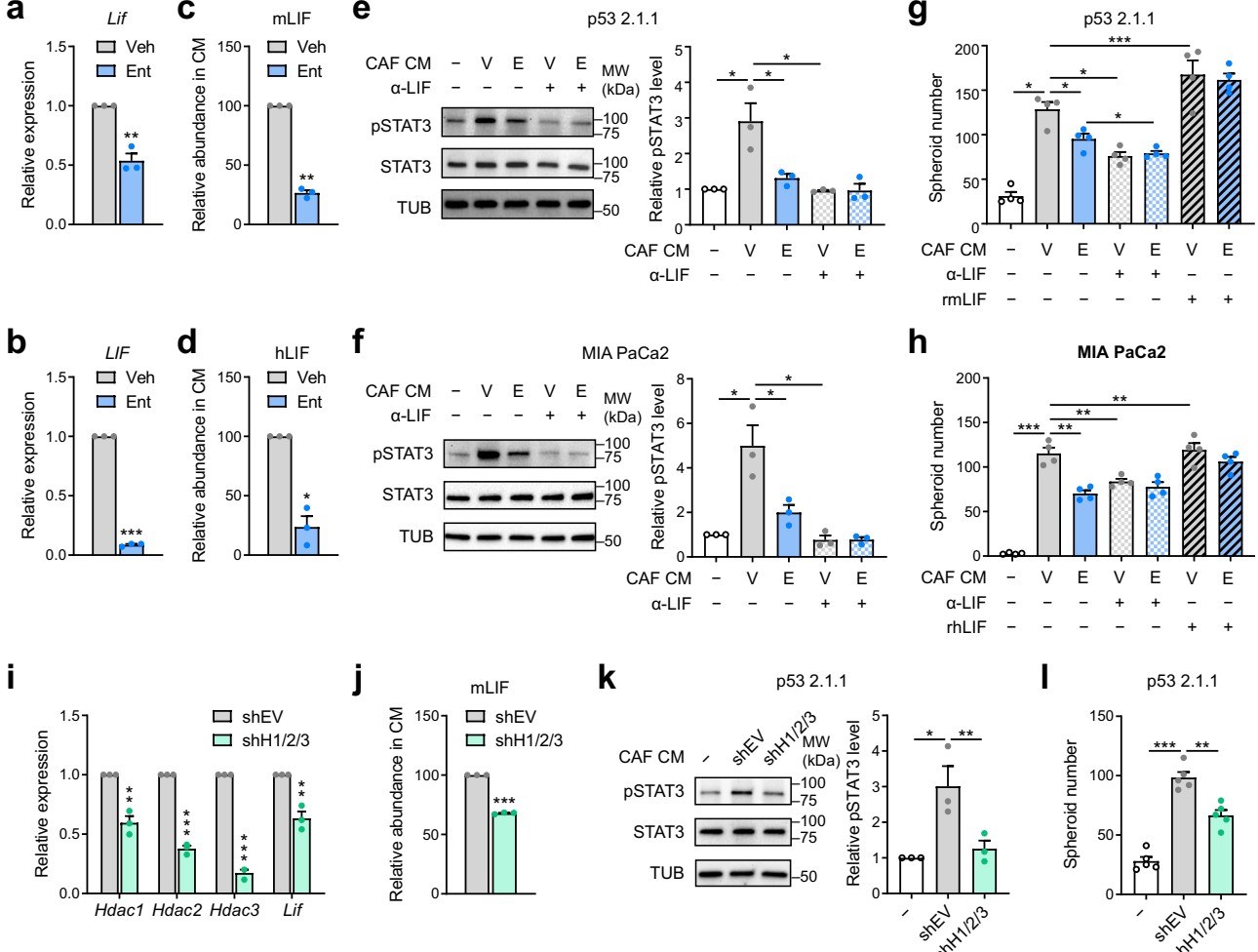

**Fig. 5 | HDACi lowers CAF-mediated pro-tumorigenic LIF-STAT3 signaling. a, b** RT-qPCR results showing *Lif*/*LIF* expression in mCAF (imCAF1, **a**) and hCAF (YAM, **b**) cells under Ent treatment (10 μM, 2 d) compared to Veh. **c, d** Relative abundance of mLIF (**c**) and hLIF (**d**) detected by immunoassay in CM from Ent-treated CAFs. **e, f** Representative images from Western blotting detecting pSTAT3, STAT3 and α-tubulin (TUB, sample processing control) in mouse (p53 2.1.1, **e**) and human PDAC cells (MIA PaCa2, **f**) treated with small molecule-depleted CM from Veh- (V) or Ent- (E) treated CAFs, and/or anti-LIF antibody (α-LIF, 4 μg/ml), with quantifications of pSTAT3/STAT3 ratio relative to no CM treatment. **g, h** Numbers of spheroids formed in p53 2.1.1 (**g**) and MIA PaCa2 cells (**h**) with CM, α-LIF antibody

(4 μg/ml) and/or recombinant (r) m/hLIF (0.1 μg/ml) treatments. **i** RT-qPCR results showing the expression of *Lif* and HDAC genes in imCAF1 cells with shRNAs targeting *Hdac1, 2* and *3* (shH1/2/3), compared to shEV. **j** Relative abundance of mLIF in CM from shH1/2/3 CAFs. **k, l** Representative images and quantifications of pSTAT3/ STAT3 ratio from Western blotting (**k**) and results from spheroid assay (**l**) in p53 2.1.1 cells treated with CM from shEV or shH1/2/3 CAFs. RT-qPCR, LIF measurements, Western blotting, *n* = 3 independent samples; spheroid assays, *n* = 4 (**g**, **h**), 5 (**l**) cell sample replicates. Data are presented as mean values ± SEM. *\*p* < 0.05; *\*\*p* < 0.01; *\*\*\*p* < 0.001. Two-sided *t*-test. Source data including *p*-values are provided as a Source Data file.

a major driver of STAT3 activation by the CAF secretome (Fig. 5e, f). Consistently, CM from Ent-treated CAFs was less efficacious in supporting spheroid formation of PDAC cells, largely replicating the effect of LIF depletion; whereas, supplementation with recombinant LIF

restored spheroid formation (Fig. 5g, h), implicating that the subpar pro-tumorigenic potential of HDAC-inhibited CAFs is due to deficient LIF secretion. Of note, residual Ent in the CM post-processing was reduced to only 30 nM (Supplementary Fig. 6c), a concentration

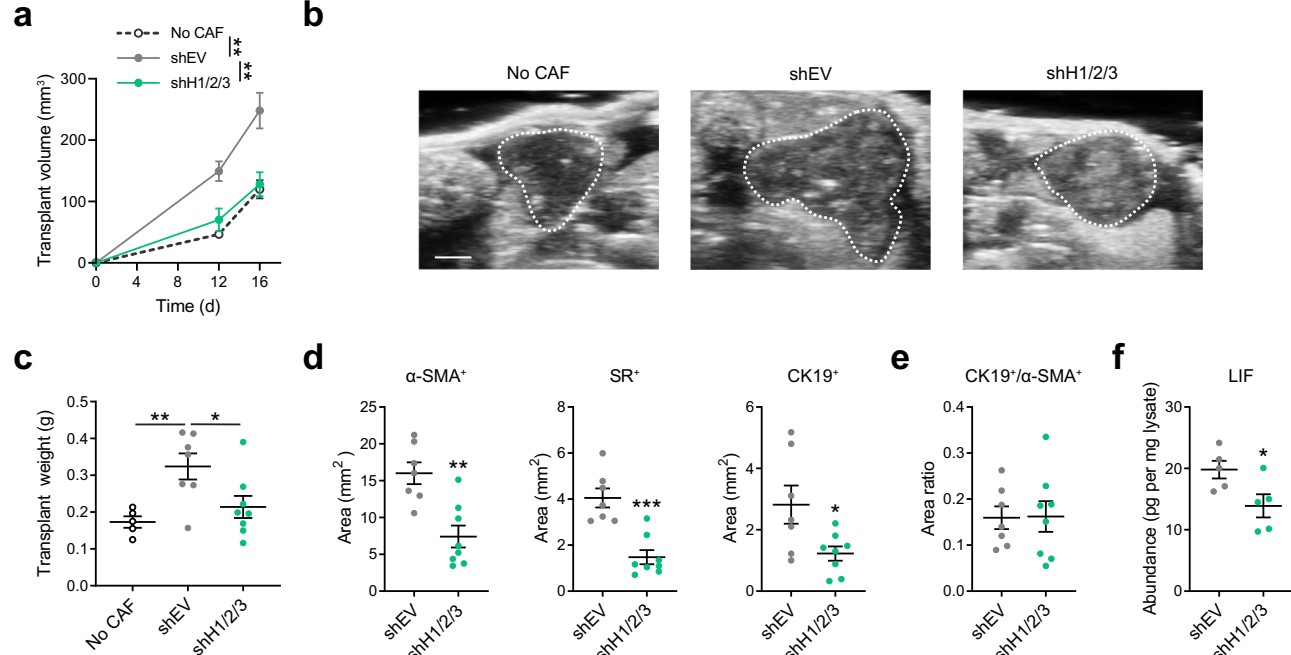

**Fig. 6 | HDAC depletion in CAFs reduces tumor progression in vivo.**
**a, b** Estimated volumes (**a**) and representative images (D16 post-transplantation, **b**) from ultrasound imaging for orthotopic transplants of PDAC cells (p53 2.1.1) with or without CAFs transduced with shEV or shH1/2/3. Tumor boundaries are highlighted with dotted lines. **c** Transplant weights at the endpoint (D19). **d, e** Measurements of total α-SMA, Sirius Red (SR) and CK19 positive areas (**d**) and ratio of CK19$^+$ and α-SMA$^+$ areas (**e**) in whole transplant sections. **f** LIF abundance per mg transplant lysates measured by immunoassay. Transplant measurements (**a, c**), $n = 5$ (no CAF), 7 (shEV), 8 (shH1/2/3) transplants; staining quantifications (**d, e**), $n = 7$ (shEV), 8 (shH1/2/3) transplants; LIF immunoassay (**f**), $n = 5$ transplant lysates. Data are presented as mean values ± SEM. *$p < 0.05$, =0.033 (**c**, shH1/2/3 vs shEV), 0.025 (**d**, CK19$^+$), 0.038 (**f**); **$p < 0.01$, =0.006 (**a**, shEV vs no CAF at D16), 0.004 (**a**, shH1/2/3 vs shEV at D16), 0.007 (**c**), 0.001 (**d**, α-SMA$^+$); ***$p < 0.001$ (**d**, SR$^+$). Two-sided $t$-test. Source data are provided as a Source Data file.

insufficient to affect STAT3 activation or spheroid formation in PDAC cells (Supplementary Fig. 6d, e). Similar to HDACi by Ent treatment, shRNA-mediated depletion of Class I HDACs (HDAC1, 2 and 3) reduced the expression and secretion of LIF (Fig. 4i, j, Supplementary Fig. 6f), Consistent with this, HDAC depletion lowered the capacity of the CAF secretome to activate tumor-intrinsic STAT3 (Fig. 5k) and to support anchor-independent growth in PDAC cells (Fig. 5l).

To further support a role for HDACs in regulating CAF pro-tumorigenicity in vivo, we performed orthotopic co-transplantation of mouse PDAC cells with HDAC-depleted CAFs and progressively monitored tumor burden. Compared to CAFs with competent HDACs, HDAC-deficient CAFs resulted in reduced tumor volume along tumor development (Fig. 6a, b), as well as lower tumor weight at the end-point (Fig. 6c). Transplants with HDAC-deficient CAFs have reduced α-SMA$^+$ fibroblast compartment and lower Sirius Red (SR)$^+$ collagen content (Fig. 6d), in agreement with the roles of HDACs in regulating stromal pro-desmoplastic programs. Notably, the cytokeratin 19 (CK19)$^+$ tumor compartment was also reduced in the transplants with HDAC-deficient CAFs (Fig. 6d), largely proportionally to the α-SMA$^+$ fibroblast compartment (Fig. 6e), indicating lower tumor-promoting effect from HDAC-deficient CAFs. In addition, lower abundance of intratumoral LIF was also detected in the transplants with HDAC-deficient CAFs (Fig. 6f), consistent with reduced tumor promotion by these CAFs. Collectively, these findings support critical roles for HDACs in regulating LIF-mediated pro-tumorigenicity in CAFs, in part by facilitating *LIF* expression and the proliferation of LIF-producing fibroblasts.

## Bicompartmental effects of HDACi reduces disease severity in PDAC models

The ability of HDACi to suppress the pro-desmoplastic and pro-tumorigenic transcriptional programs in CAFs/PSCs suggests that

HDACi may have therapeutic utility as a stromal targeting approach. To fully delineate the therapeutic potential of HDACi, the impacts of Ent on pancreatic tumor cells were also evaluated. Ent treatment reduced tumor cell numbers in cultures of human and mouse PDAC cell lines and organoids (IC$_{50}$ 10$^{-5}$ to 10$^{-6}$ M) (Supplementary Fig. 7a, b); effects largely attributed to cell cycle arrest (Supplementary Fig. 7c, d) rather than apoptosis (Supplementary Fig. 7e). Depletion of HDAC1, but not HDAC2 or HDAC3, in PDAC cells largely replicated this pro-liferative defect (Supplementary Fig. 7f). Furthermore, Ent treatment downregulated genes important for cell cycle progression in mouse and human PDAC cells (Supplementary Fig. 7g, h), consistent with the observed anti-proliferative effects. Interestingly, multiple core biolo-gical pathways, including RNA processing, transcription, DNA repli-cation and damage repair, were also downregulated by Ent (Supplementary Fig. 7i), implicating a role of HDACs in coordinating the transcriptional programs important for neoplastic transformation. In contrast, Ent upregulated many epithelial markers, as well as other genes in the pathways essential for epithelium functions, including angiogenesis, wound healing, epithelium development and differ-entiation (Supplementary Fig. 3g–i). These data, along with previous publications[31,32,45], evidence the tumor-targeting capacity of HDACi.

To evaluate the efficacy of HDACi in vivo, Ent treatment was carried out in $KP^{f/f}C$ mice ($Kras^{LSL-G12D/+};Trp53^{f/f};Pdx1-Cre$), a genetically engineered mouse model (GEMM) in which PDAC progresses in a relatively synchronous manner[9]. Ent monotherapy of $KP^{f/f}C$ mice significantly reduced tumor burden (41%) (Fig. 7a, b) and extended survival (35%) (Fig. 7c). Histopathological analysis revealed that Ent treatment reduced the cases of high-grade poorly differentiated tumors (Fig. 7d, e), implicating the capacity of HDACi to arrest tumor progression. In addition, Ent treatment reduced the volumes of the CK19$^+$ tumor cells, as well as the α-SMA$^+$ activated stromal fibroblasts and the SR$^+$ stromal collagen content (Fig. 7f, Supplementary Fig. 8a),

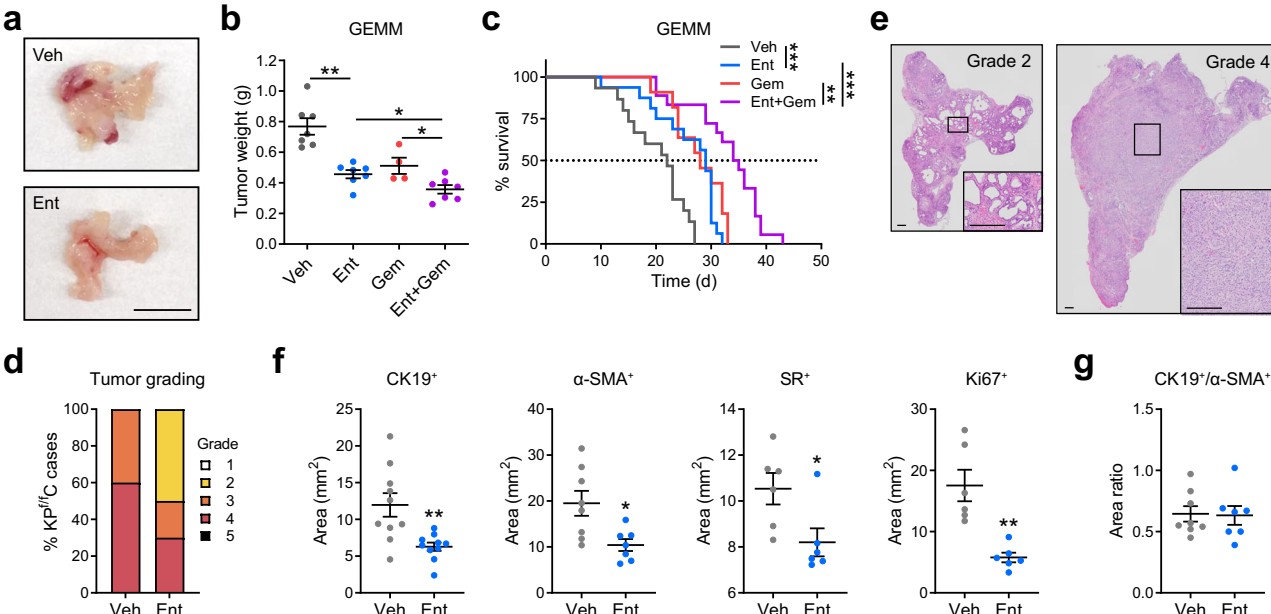

**Fig. 7 | Ent treatment arrests tumor progression and provides therapeutic benefits in PDAC GEMM. a, b** Representative tumor images (**a**) and tumor weight measurement (**b**) from GEMM (*KP^{f/f}C* mice) after 3-week treatment of Veh, Ent (5 mg/kg, daily), Gem (25 mg/kg, q3d) or the combination of Ent and Gem. Scale bar, 10 mm. *n* = 7 (Veh, Ent, Ent+Gem), 4 (Gem) mice. **c** Kaplan-Meier curves showing the survival time after treatment initiation in *KP^{f/f}C* mice. *n* = 15 (Veh), 16 (Ent), 11 (Gem), 18 (Ent+Gem) mice. Median survival (d): 22 (Veh), 29 (Ent), 28 (Gem), 34.5 (Ent+Gem). **d, e** Pathological grading of tumor samples from Veh- or Ent-treated *KP^{f/f}C* at moribund (**d**), and representative images of Grades 2 (well differentiated) and 4 (poorly differentiated) tumor sections with hematoxylin and eosin (H&E) staining (**e**). Scale bar, 25 μm. *n* = 10 tumors. **f, g** Measurements of total CK19, α-SMA, and Ki67 and SR positive areas (**f**) and ratio of CK19⁺ over α-SMA⁺ areas (**g**) in the whole tumor sections from *KP^{f/f}C* mice under Veh or Ent treatment. *n* = 10 (**f**, CK19⁺), 8 (Veh in **f**, α-SMA⁺ and **g**), 7 (Ent in **f**, α-SMA⁺ and **g**), 6 (**f**, SR⁺ and Ki67⁺). *p* = 0.904 (**g**). Data in **b**, **f**, **g** are presented as mean values ± SEM. *\*p* < 0.05, =0.025 (**b**, Ent+Gem vs Ent), 0.019 (**b**, Ent+Gem vs Gem), 0.013 (**f**, α-SMA⁺), 0.029 (**f**, SR⁺); *\*\*p* < 0.01, =0.002 (**c**, Gem vs Ent+Gem), 0.004 (**f**, CK19⁺), 0.001 (**f**, Ki67⁺); *\*\*\*p* < 0.001 (**b**, Ent vs Veh; **c**, Ent vs Veh; **c**, Ent+Gem vs Ent). Survival analysis (**c**), log-rank test; others (**b**, **f**), two-sided *t*-test. Source data are provided as a Source Data file.

supporting the bicompartmental targeting capacities of Ent to reduce tumor progression and stromal activation. Fewer Ki67⁺ proliferative cells were also observed under Ent treatment (Fig. 7f, Supplementary Fig. 8a), consistent with the ability of Ent to induce cytostasis in both compartments. Notably, the tumor/myofibroblast ratio (CK19⁺/α-SMA⁺) was maintained (Fig. 7g), which together with reduced tumor aggressiveness (Fig. 7d, f) suggests that Ent treatment avoids the detrimental effects associated with near-complete stromal fibroblast depletion[19,20]. Moreover, Ent treatment increased the presence of CD8⁺ T cells in the TME (Supplementary Fig. 8b, c), implicating alleviated immunosuppression under HDACi treatment[46,47]. In addition, combined treatment of Ent with gemcitabine (Gem), a front-line chemotherapeutic agent frequently used in PDAC patients, enhanced the therapeutic benefits in *KP^{f/f}C* mice with a 53% reduction in tumor burden and a 60% increase in survival compared to vehicle treatment (Fig. 7b, c). In addition, tumor reduction by Ent was also seen in syngeneic orthotopic transplantation and patient-derived xenograft (PDX) models (Supplementary Fig. 8d, e), further supporting the potential of HDACi to improve clinical outcomes. Collectively, these data demonstrate the therapeutic efficacy of the HDACi-based bicompartmental targeting approach.

## HDACi alters stromal fibroblast heterogeneity in vivo

Along with reduced PDAC severity, the concurrent reduction in α-SMA⁺ activated stromal fibroblasts under Ent treatment (Figs. 5d, 6f) implicates HDAC activities in regulating stromal fibroblast heterogeneity. To shed light on the impact of HDACi on fibroblast composition, the single-cell transcriptome changes induced by Ent were determined in fibroblasts isolated from *KP^{f/f}C* mouse tumors via marker PDPN (podoplanin) (Supplementary Fig. 9a). Among the 10 fibroblast subpopulations identified, Subpopulation 1–5 show high

expression of markers identified in inflammatory or adventitial fibroblasts, such as *Ly6c1* (the major coding gene for Ly6C surface antigen) and others[24,25], but low expression of myofibroblast markers like *Acta2* and *Tagln* (Fig. 8a–c, Supplementary Fig. 9b, Supplementary Data 3) (hereafter, referred as *Ly6c1^{Hi}* subpopulations); in contrast, Subpopulations 6–10 (referred as *Ly6c1^{Lo}* subpopulations) show low expression of inflammatory/adventitial markers but high expression of myofibroblast markers, and includes typical myofibroblastic populations[13,24,25], a proliferative subpopulation (Subpopulation 10, marked by *Mki67*) and a subpopulation with antigen-presenting potential (Subpopulation 6, marked by *Cd74*)[24] (Fig. 8a–c, Supplementary Fig. 9b, Supplementary Data 3). Remarkably, Ent treatment reduced the frequency of Subpopulation 8 and 9, two major myofibroblastic subpopulations; meanwhile, it significantly enriched Subpopulation 3, one of the *Ly6c1^{Hi}* subpopulations (Fig. 8d–f). This subpopulation features lipid-related genes as markers (e.g. *Fabp4*, *Lcn2*) (Fig. 8g, Supplementary Fig. 9b, Supplementary Data 3) and displays a trend of high expression of lipogenic transcriptional regulators including PPAR-γ (peroxisome proliferator-activated receptor γ, gene name *Pparg*) and PGC1-α (PPAR-γ coactivator α, gene name *Ppargc1a*; Supplementary Fig. 9c), implicating the lipogenic potential in these fibroblasts. Similar subpopulations highly expressing lipogenic markers were also detected in *Ly6c1^{Hi}* fibroblasts from other datasets of PDAC-associated fibroblasts (Supplementary Fig. 9d, e)[24,25]. The capacity of Ent to increase the ratio of lipogenic to myofibroblastic subpopulation indicates the lipogenic subpopulation as potential myofibroblast precursors. Indeed, trajectory analysis suggested the lipogenic subpopulation may be one of the immediate precursor populations prior to myofibroblast activation (Supplementary Fig. 9f). Furthermore, the ability of Ent treatment to upregulate lipogenic markers and downregulate myofibroblast markers in most fibroblast

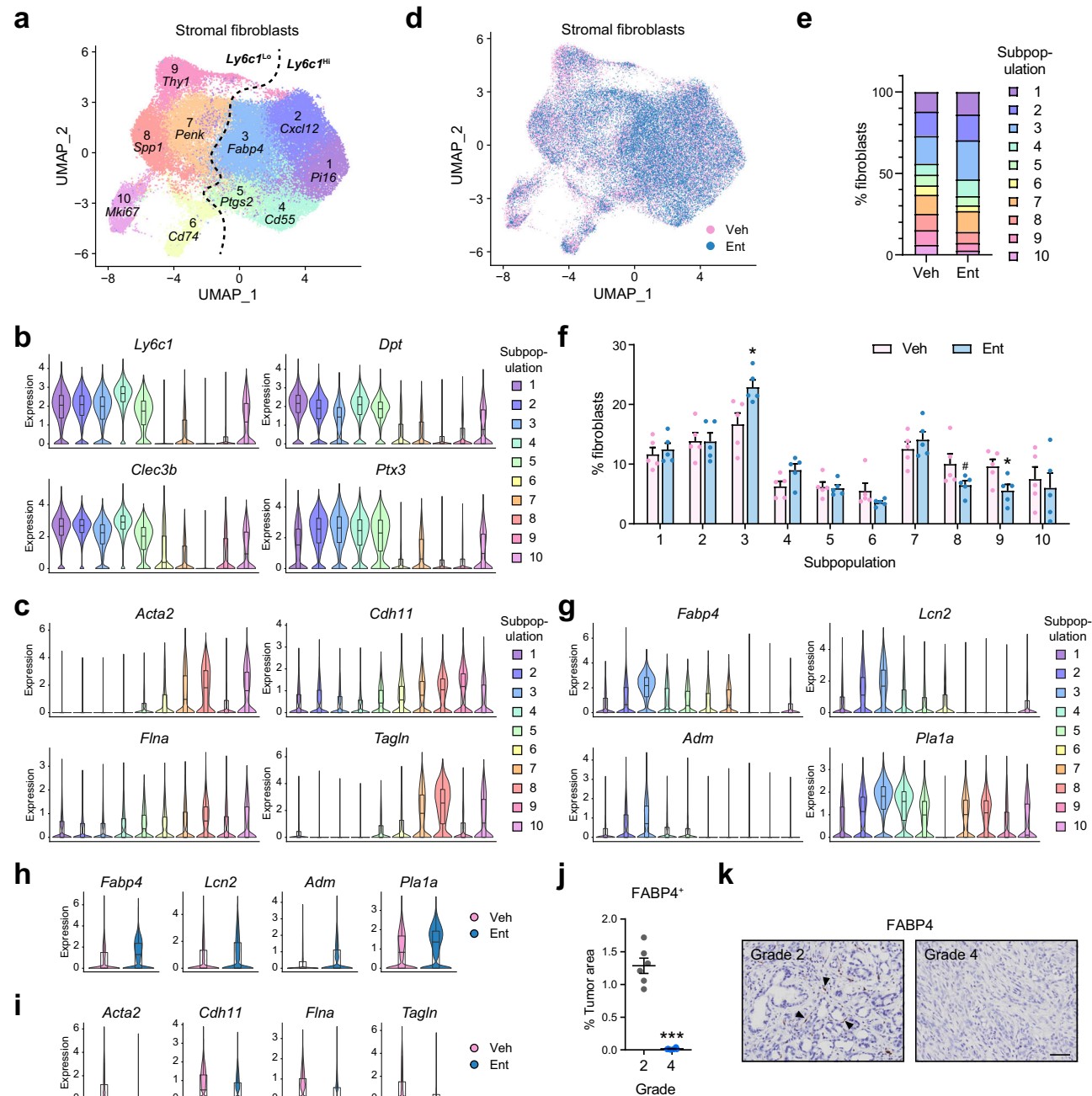

**Fig. 8 | Ent treatment enriches lipogenic fibroblasts and reduced myofibroblast frequency in vivo. a** UMAP showing 10 stromal fibroblast subpopulations with representative markers from tumors in $KP^{f/f}C$ mice. $n$ = 56,723 cells from 10 mice. **b, c** Violin plots showing the expression of selected inflammatory/adventitial (**b**) and myofibroblastic markers (**c**) in fibroblast subpopulations. **d** UMAP showing the distribution of stromal fibroblast subpopulations from Veh- or Ent-treated $KP^{f/f}C$ mice. $n$ = 30,867 cells from 5 Veh-treated mice, 25,826 cells from 5 Ent-treated mice. **e, f** Bar graphs showing the percentages of fibroblast subpopulations on average (**e**) and in individual mice (**f**). #, $p$ = 0.062 (Subpopulation 8); *$p$ < 0.05, =0.001 (Subpopulation 3), 0.033 (Subpopulation 9). Two-sided $t$-test. **g** Violin plots showing the

expression of selected markers for the lipogenic subpopulation (Subpopulation 3). **h, i** Violin plots showing that Ent enhances the expression of lipogenic markers (**h**) and reduces myofibroblast markers (**i**) in the bulk fibroblast populations. **j, k** Quantifications (**j**) and representative images (**k**) of FABP4 staining in representative areas of Grades 2 and 4 tumors from $KP^{f/f}C$ mice. $n$ = 6 representative areas. ***$p$ < 0.001. Two-sided $t$-test. Scale bar, 200 µm. Boxes in violin plots (**b**, **c**, **g**, **h**, **i**): center lines, medians; box limits, first and third quartiles; whiskers, minima (first quartiles −1.5 × IQR) and maxima (third quartiles + 1.5 × IQR). IQR, interquartile range. Data in **f, j** are presented as mean values ± SEM. Source data are provided as a Source Data file.

subpopulations translated into detectable transcriptional changes in the total fibroblast population (Fig. 8h, i, Supplementary Fig. 9g–l). Notably, the expression of pro-desmoplastic TF gene *Srf*, which is mostly limited to myofibroblastic/proliferative subpopulations, was also reduced by Ent treatment (Supplementary Fig. 9j), as well as the TF gene *Foxm1* and the proliferation genes (Supplementary Fig. 9m), in agreement with Ent's capacity to block the pro-desmoplastic

transcriptional programs driven by these TFs. In addition, the fact that Class I HDACs were poorly expressed in the lipogenic and other $Ly6c1^{Hi}$ subpopulations but highly expressed in myofibroblastic/proliferative subpopulations (Supplementary Fig. 9n) suggests key roles for HDACs in driving stromal activation, and the reduced expression of HDAC genes under Ent treatment (Supplementary Fig. 9n) implicates a positive feedback loop underlying the inhibitor's effects.

Furthermore, the frequent detection of FABP4+ lipogenic fibroblast-like cells in low-grade well differentiated tumors, plus their absence in high-grade poorly differentiated tumors (Fig. 8j, k), associates lipogenic fibroblasts with benign tumors and less activated stroma, consistent with the finding that FABP4 is one of the top stromal markers that predict good patient prognosis[17]. The increased occurrence of low-grade tumors under Ent treatment (Fig. 7d) also agrees with the enhanced frequency of lipogenic fibroblasts by Ent (Fig. 8d–f), implying that the HDAC-regulated switch of stromal transcriptional programs is highly coordinated with tumor progression. Together, the above findings demonstrate that HDACi treatment blocks the lipogenic-to-myofibroblastic program switch in vivo, reversing the fibroblast composition change in stromal activation and tumor progression.

## Correlation of HDAC expression and patient prognosis

Our findings associate HDAC activities with stromal activation and PDAC progression. To explore the relevance of these findings to human disease, we analyzed HDAC expression in patient cohorts in publicly available datasets. In a patient cohort stratified by normal and activated stroma, which associate with good and poor prognosis respectively[17], increased *HDAC1* expression was detected in PDAC samples with activated stroma (Supplementary Fig. 10a), implicating the contribution of stromal HDAC activities to disease progression. Moreover, in another patient cohort in which gene expression was analyzed with bulk tumors[48], high *HDAC1* expression signal also correlates with poor prognosis in PDAC patients (Supplementary Fig. 10b, c), in agreement with the associations of PDAC prognosis with both high tumor[33,34] and high stromal HDAC activities (Supplementary Fig. 10a). As HDAC1/2 chromatin actions are mediated by multi-component protein complexes (Supplementary Fig. 10d), we examined whether the expression of HDAC1/2-containing complex components similarly correlate with patient prognosis. Indeed, the expression of components of the NuRD, SIN3 and CoREST complexes provided superior prognostic power to distinguish high and low-risk patients compared to *HDAC1* expression alone (Supplementary Fig. 10e). Overall, the analysis of patient data links patient prognosis with stromal and bulk tumor HDAC expression.

## Discussion

The poor prognosis of PDAC is a direct consequence of the dearth of effective therapies, and indirectly, of our incomplete understanding of how tumor aggressiveness is promoted by an intricate network of molecular and cellular components in the TME. Here we report key roles for Class I HDACs in inducing the transcriptional programs that facilitate stromal activation and tumor promotion, and establish HDACi as an effective stromal targeting strategy (Supplementary Fig. 10f). We show that HDACs facilitate stromal fibroblasts to switch from transcriptional programs associated with normal physiology (e.g., a lipogenic program) to the pro-desmoplastic programs driving stromal activation and supporting tumor progression (Figs. 1, 4, 8). Furthermore, the ability of HDACs to mediate transcriptional programs responsive to pro-inflammatory/pro-tumorigenic signals (e.g. TNF-α) contributes to LIF-mediated pro-tumorigenicity in CAFs (Figs. 4d, e, j–m, 5). The effects of HDACi on fibroblast transcriptional programs result in suppressed stromal activation and pro-tumorigenicity contributing to reduced disease severity in mouse models (Figs. 6, 7, Supplementary Fig. 8). The fact that stromal HDAC expression predicts poor prognosis in PDAC patients (Supplementary Fig. 10a) further supports the essential roles of stromal HDACs in contributing to PDAC progression.

The ability of HDACi to interrupt the establishment of activation-specific chromatin architecture identifies Class I HDACs as important effectors coordinating chromatin accessibility with transcriptional activities required for PSC activation (Fig. 3). The

molecular details on how HDACs maintain chromatin architecture in PSCs remain to be further examined. In addition, we identify FOXM1 and SRF as a hierarchical regulatory axis mediating the HDAC-coordinated pro-desmoplastic transcriptional programs, with SRF directing the proliferation program through FOXM1 and the myofibroblast program by itself or through other downstream TFs (Fig. 2), revealing the transcriptional logics underlying PSC/CAF activation.

Concurrent with the suppression of pro-desmoplastic programs is the re-engagement of the lipogenic transcriptional program (Figs. 1d, e, g, j, 4a, d, 8). The fact that lipogenic fibroblasts were detected in benign stage PDAC but depleted in the advanced stage (Fig. 8) is consistent with the association of lipid features with pre-activated/normal pancreatic stromal fibroblasts such as PSCs[3,17]. Given the potential of Ent to block PSC/CAF activation, the shifted composition of lipogenic and myofibroblastic fibroblasts under Ent treatment (Fig. 8) implicates the lipogenic population as presumptive fibroblast precursors in the TME prior to myofibroblast activation. Additional studies are needed to further characterize the functional role of lipogenic fibroblasts, as well as the dynamics of fibroblast subpopulations during tumor progression.

Beyond regulating the pro-desmoplastic and lipogenic programs, HDACs also coordinate a pro-inflammatory transcriptional program inducible by environmental signals like TNF-α (Fig. 4d, e, j–m). One of the key effectors under this program is LIF, a pro-tumorigenic cytokine promoting PDAC progression and chemoresistance[9]. The reduction of LIF production in CAF by Ent lowered the pro-tumorigenic crosstalk mediated by LIF and the tumor LIFR-STAT3 pathway (Fig. 5), phenocopying the effect of LIF blockade and potentially contributing to tumor progression arrest and sensitization to chemotherapy. Of note, HDACi has been shown to enhance the secretion/expression of certain inflammation-related factors (e.g. IL8) in CAFs[49]; however, the functional significance of such an HDACi-induced secretome remains to be established.

In addition to the activities in PSCs/CAFs, Class I HDACs in pancreatic tumor cells potentiate the switch from the transcription programs supporting epithelial physiological functions to the programs driving neoplastic transformation (Supplementary Fig. 7). The bicompartmental targeting potentials of HDACi arrest tumor progression and improve therapeutic outcomes as observed in the mouse models (Fig. 6, Supplementary Fig. 8). Along with recent findings that HDAC inhibitors sensitize tumors to immune checkpoint blockades[46,47,50,51] and other epigenetic agents[28], our data highlight the potential of this class of epigenetic modulators in PDAC therapeutics. Strategies to boost the efficacy of HDACi should be developed in a quest for successful clinical translation. Overall, our study uncovers important roles of HDACs in regulating the transcriptional programs driving stromal activation and pro-tumorigenicity and provides the scientific foundation for using HDACi as a potent stromal targeting strategy.

## Methods

All procedures were conducted in compliance with relevant institutional and national ethical guidelines. All animal protocols were approved by the Institute of Animal Care and Use Committee (IACUC) at the Salk Institute, and the University of Arizona (in which animal studies of the Translational Genomic Research Institute are performed).

### Cell culture

Primary mouse PSCs were isolated at use from wild-type C57BL/6J male mice at the age of 8–12 weeks. The mouse CAF lines imCAF1 and imCAF2 were derived from primary CAFs from *Kras*$^{LSL-G12D/+}$;*Trp53*$^{f/f}$; *Pdx1-Cre* (*KP*$^{f/f}$*C*) mice (FVB/NJ), isolated by FACS sorting via marker Pdpn and immortalized by lentivirus expressing SV40 large T antigen

with RFP (GenTarget, LVP016-RB). The patient-derived primary CAF ONO and immortalized PSC/CAF lines YAM and hPSC1 were provided by Atsushi Masamune (Tohoku University). The mouse PDAC cell line FC1245 was derived from a *Kras^LSL-G12D/+;Trp53^LSL-R172H/+;Pdx1-Cre* (*KPC*) mouse, (C57BL/6J) and provided by David Tuveson (Cold Spring Harbor Laboratory); p53 2.1.1 was derived from a *Kras^LSL-G12D/+;Trp53^f/+;Ptf1a-Cre* mouse (FVB/NJ) and provided by Eric Collison (University of California, San Francisco). The human PDAC cell lines MIA PaCa2, PSN1 and Panc1 were acquired from ATCC. The GFP-expressing FC1245 line was derived from a single cell clone of FC1245 cells transduced with lentivirus expressing GFP. 3D culture of CAFs were performed with Matrigel (Corning). All mouse and human PSCs, CAFs and PDAC cells were cultured in DMEM (Corning) with 10% fetus bovine serum (FBS, characterized, HyClone) and 1× Antibiotic/Antimycotic (100 units/ml penicillin, 100 μg/ml streptomycin and 250 ng/ml amphotericin B; Gibco) except specified otherwise.

## PDAC organoid culture
The mouse PDAC organoid mT9 was derived from a *KPC* mouse (C57BL/6J)[52] and cultured in Matrigel with Mouse Complete Feeding Medium (Advanced DMEM/F12 with 10 mM HEPES, 1× Glutamax, 500 nM A83-01, 50 ng/ml hEGF, 100 ng/ml mNoggin, 100 ng/ml hFGF10, 0.01 μM hGastrin I, 1.25 mM N-acetylcysteine, 10 mM nicotinamide, 1× B27 supplement and 10% R-spondin1 conditioned media) with penicillin (100 units/ml) and streptomycin (100 μg/ml). The human PDAC organoids hF3, hM1A, hT3 were cultured in Matrigel with Human Complete Feeding Medium (Mouse Complete Feeding Medium plus 1 μM PGE2, and 50% afamin/Wnt3A conditioned media) with penicillin and streptomycin[52,53]. All the PDAC organoids were provided by David Tuveson (Cold Spring Harbor Laboratory) and Hervé Tiriac (University of California, San Diego).

## Animals
Mice were housed at a temperature ($22 \pm 1$ °C) and humidity (45–65%) controlled environment with a 12-h light-dark cycle. Wild-type C57BL/6J male mice (8–12 weeks, the Jackson Laboratory) were used as PSC donors and hosts of orthotopic transplantation with FC1245 cells. Wild-type FVB/NJ male mice (9–10 weeks, the Jackson Laboratory) were used as hosts of orthotopic co-transplantation of CAFs (imCAF1) and PDAC cells (p53 2.1.1). *KP^f/f^C* mice with *Rosa26^luc/luc* (female and male) in the FVB/NJ background were used as a GEMM[9], and both female and male were enrolled in therapeutic treatment around the age of 25. Athymic nude mice (female, 4–6 week, Taconic) were used as hosts of subcutaneous implantation of PDX, and sacrificed when tumors were beyond 2000 mm³. Mouse euthanasia was performed with CO2 exposure followed by cervical dislocation. No mice in the study carried tumors exceeding 2000 mm³, the tumor size limit approved by IACUC.

## Mouse PSC isolation and in vitro activation
Isolation of PSCs from pancreata of wild-type C57BL/6J mice was carried out with a protocol adapted from previous reports[6,54]. Briefly, pancreatic tissues from mice were incubated in Gey's balanced salt solution (GBSS; Sigma-Aldrich) with 0.05% Collagenase P (Roche), 0.02% Pronase (Roche, Indianapolis, IN) and 0.1% DNase I (Roche) at 37 °C with agitation for 20 min. Digested tissue was filtered through a 100 mm nylon mesh. Cells were subsequently pelleted, washed once with GBSS, and resuspended in 9.5 ml GBSS containing 0.3% bovine serum albumin (BSA) and 8 ml 28.7% Histodenz solution (Sigma-Aldrich) to reach an approximate density of 1.070. The cell suspension was layered beneath GBSS containing 0.3% BSA, and centrifuged at 1400 g for 20 min at 4 °C. PSCs were retrieved at the interface of aqueous and Histodenz-containing solution. Harvested PSCs were washed with GBSS, resuspended in DMEM with 10% FBS and 1× Antibiotic/Antimycotic and seeded on tissue culture plates. PSCs were

subjected to culture-induced activation for no more than 6 d. PSCs cultured for <24 h were used as pre-activated PSCs; those cultured for 3 d and beyond were used as activated PSCs.

## Drug preparation and administration
Ent was synthesized by WuXi AppTec; Gem was purchased from Torcis (3259). For in vitro experiments, Ent was dissolved in dimethylsulfoxide (DMSO) at 10 mM, stored at −20 °C, and diluted to specified concentrations at use. The dosages of Ent used in PSCs, CAFs and PDAC cells in vitro were determined by preliminary experiments and selected for effective dosages around IC$_{50}$s (0.1–1×) in functional assays. The treatment durations were also optimized for manifesting the functional drug effects. For mouse administration, Ent was dissolved in phosphate-buffered saline (PBS) with 0.05 N HCl and 0.1% Tween-20 at 5 mg/ml, diluted to 1 mg/ml for oral administration (p.o.) and stored at 4 °C for up to 3 months; Gem was dissolved in sterile saline at 5 mg/ml for i.p. and stored at −20 °C. The in vivo dosing schemes were optimized by pilot experiments, and the applied dosages were selected at or below the maximum tolerated dosages in respective animal models.

## In vitro proliferation/viability assay
For the proliferation/viability assays on PSCs, ex vivo PSCs after 1 d culture were seeded into 384-well plates with 50 cells per well and applied with Ent at 0.01–100 μM using a D300e digital dispenser (HP). After 2 or 5 d Ent treatment, CellTiter-Glo assay (Promega) was performed and luminescent signals measured by an Envision plate reader (Perkin Elmer). For the assays on CAFs, imCAF1 and imCAF2 were seeded into 384-well plates with 150 cells per well, 24 h before Ent treatment (0.01–100 μM). CAF cultures were treated for 2 d, before CellTiter-Glo assay. For the assays on PDAC cells, mouse cell lines FC1245 and p53 2.1.1 and human lines MIA PaCa2, Panc1, and PSN1 were seeded into 384-well plates at 150, 120, 210, 420 and 210 cells per well, respectively. 24 h after seeding, cultures were treated with Ent (0.01–100 μM) for 2 (p53 2.1.1) or 3 d (others) before CellTiter-Glo assay. For the assays on PDAC organoids, mT9, hM1A, and hT3 organoids were dissociated into single cells and seeded at 500 cells per well in 384-well plates in 30 μl of Mouse Complete Feeding Medium with 10% Matrigel. 24 h after seeding, cells were treated with Ent at 0.05–50 μM for 3 d, following by CellTiter-Glo assay. Relative cell numbers were calculated by comparing luminescence signals from drug treatments to those from no drug treatments.

## Flow cytometry analysis
Ent treatments were performed with PSCs (D1) for 5 d at 5 μM, imCAF1 cells for 2 d at 10 μM, and FC1245 and MIA PaCa2 cells for 2 d at 5 μM, before apoptosis and/or cell cycle distribution analyses via flow cytometry. For apoptosis analysis, staining was performed using Annexin V detection kit (APC; eBiosciences, 88-8007-72) with 7-AAD (eBiosciences, 00-6993-50). For cell cycle distribution analysis with EdU labeling, cells were incubated with EdU (100 μM, 1 h), dissociated for single cells, and processed using Click-IT EdU assay kit (Alexa Fluor 647; Invitrogen, C10424) with DNA dye Hoechst 33342 (10 μg/ml). Data were collected on a FACSCantoII or LSR-II flow cytometry (BD), and analyzed with FlowJo software (v10.7.1).

## RNA-seq analysis
Total RNA was isolated by RNeasy Mini or RNeasy Plus Micro Kit and controlled for quality with a 2100 Bioanalyzer (Agilent) prior to cDNA libraries construction. Libraries were prepared using 100–500 ng total RNA with TruSeq Stranded RNA Sample Preparation Kit (Illumina, v2) according to the manufacturer's protocol. Briefly, mRNA was purified, fragmented, and used for first-, then second-strand cDNA synthesis followed by adenylation of 3' ends. Samples were ligated to unique adapters and subjected to PCR amplification.

Libraries were then validated by BioAnalyzer, normalized and pooled for sequencing. High-throughput single-end sequencing was performed on the HiSeq 2500 system (Illumina) with a 100-bp read length. Image analysis and base calling were performed with CASAVA (Illumina, v1.8.2). Short read sequences were mapped to mouse (GRCm38) or human reference genomes (GRCh37) using the RNA-seq aligner STAR (v2.5.1b)[55]. Known splice junctions from mouse (Ensembl) or human (UCSC) genome annotations were supplied to the aligner and de novo junction discovery was also permitted. Differential gene expression analysis and statistical testing were performed using Cuffdiff 2 (v2.2.1)[56]. Transcript expression was calculated as gene-level relative abundance in fragments per kilobase of exon model per million mapped fragments (FPKM) and employed correction for transcript abundance bias. Hierarchical clustering was carried out using Cluster software (v3.0) and visualized by Java Treeview (v3.0).

### RT-qPCR
Total RNA (50–100 ng) was used for cDNA synthesis with iScript reagent (Bio-Rad). qPCR was performed using a CFX384 detection system (Bio-Rad) by mixing cDNAs, gene-specific primers and SsoAdvanced SYBR Green reagent (Bio-Rad). Expression was analyzed using CFX Maestro 2.3 software (Bio-Rad, v5.3.022.1030) with normalization to *Rplp0* (mouse) or *SNORD36B* (human) expression. Primer sequences are described in Supplementary Data 4.

### GSEA, GO and TF-binding enrichment analysis
GSEA analysis was performed using GSEA software (Broad Institute, v4.0.3) and run with gene set permutation, log2 ratio of classes as ranking matric and other default parameters[57]. GO analysis was performed using the database DAVID (v6.8) with default parameters[58]. TF binding enrichment analysis was performed using Enrichr[59,60] and results from databases of ChEA and ENCODE were reported.

### Immunofluorescence microscopy
PSCs were seeded on Millicell EZ slides (Millipore) 24 h before being processed for immunofluorescence staining. EdU labeling was performed with Click-iT EdU image kit (Alexa Fluor 488; Invitrogen, C10337) with 1 h EdU (10 μM) incubation. Staining was performed using a standard protocol with primary antibodies against α-SMA (Santa Cruz, sc-32251, 1:100) and Ki67 (Abcam, ab15580, 1:500) and fluorochrome-conjugated secondary antibodies against mouse (Alexa Fluor Plus 555; Invitrogen, A32727, 1:2000) and rabbit IgG (Alexa Fluor 647; Invitrogen, A27040, 1:2000), BODIPY 493/503 (10 μg/ml), and Hoechst 33342 (10 μg/ml). Mounted slides were imaged with an LSM 710 laser scanning confocal microscope system (Zeiss) or a VS-120-L100 virtual slide system (Olympus).

### Lentiviral shRNA production
The seed shRNA sequences were designed by BLOCK-iT RNAi Designer (ThermoFisher) and cloned into lentiviral expressing vector pTY-U6-Pgk-Puro. 3–5 candidate sequences were tested for each target gene, and candidates with the best inhibition efficiency and without any apparent off-target effects were selected for further experiments. The seed sequences selected were 5'-GCCTGCACCATGCAAAGAAGT-3' for *Hdac1*, 5'-GCCAAGAAGTCAGAAGCATCA-3' for *Hdac2*, 5'-GCCGCTAC-TATTGTCTCAATG-3' for *Hdac3*, 5'-GCAAATTTCCAGCCGGAATCA-3' for *Foxm1* and 5'-GGAAGACGGGCATCATGAAGA-3' for *Srf*. Individual lentiviral constructs were transfected into 293T cells along with plasmids pHP, HEF1-VSVG and pCEP4-Tat. 5 h after transfection, transfected cells were cultured in FreeStyle 293 Expression Medium (Gibco, 12338018) for a total of 48 h, with supernatant harvested every 24 h. Virus-containing supernatants were concentrated and tested for infection efficiency before transduction. Transduced cells were subjected to puromycin selection (2 μg/ml) for 2 d.

### ATAC-seq analysis
PSC samples, each with $5 \times 10^4$ cells, were harvested, washed with PBS and incubated with lysis buffer (10 mM Tris-Cl [pH 7.4], 10 mM NaCl, 3 mM MgCl$_2$, and 0.05% NP-40) for 10 min. Nuclei were pelleted and incubated with Tn5 Transposase (Nextera kit; Illumina) for 30 min at 37 °C with constant agitation. Fragmented DNA resulting from the transposition reaction was purified using PCR MinElute purification kit (Qiagen), followed by amplification with indexed primers[61]. The amplified libraries were sequenced as paired-end reads using NextSeq 500 System (Illumina). Reads were mapped to mouse reference genome (mm9, UCSC) using Bowtie2 (v2.3.4.3) with default parameters (e.g. -X 500, -sensitive)[62]. PCR duplicates and mitochondrial reads were removed. Peak calling, gene annotation and motif analysis were conducted with HOMER software (UCSD, v4.0) with default settings[63]. The most adjacent method was used for annotating peaks to genes with the maximum distance up to 2 Mb. The ± 200 bp region from the peak center was used for motif finding.

### Cytokine stimulations in CAFs
TGF-β stimulation was performed with TGF-β1 (R & D Systems, 100-B-001) at 1 ng/ml for 48 h in primary human CAFs ONO co-treated with Veh or Ent (10 μM). TNF-α stimulation was performed in immortalized CAF line YAM, which were conditioned with DMEM with 2% FBS and 1× Antibiotic/Antimycotic for 24 h. Conditioned cells were then subjected to Veh or Ent (10 μM) treatment for 40 h, following by stimulation of human TNFα (10 ng/ml; Peprotech, 300-01 A) for 8 h.

### Immunoassays for LIF detection
Immunoassays were performed with 25 μl of CM or tissue lysate each reaction, using a Bio-Plex 200 System (Bio-Rad) with Milliplex Map Mouse Cytokine/Chemokine Panel I (Millipore, MCYTOMAG-70K) for detecting mLIF or Bio-Plex Pro Human Cytokine LIF Set (Bio-Rad, 171B6011M) for detecting hLIF. Fluorescence acquisition and concentration calculation were carried out with a Bio-Plex 200 system (Bio-Rad). Quantifications were performed with at least three independent samples, each of which was represented by the average of two technical replicates.

### Preparation of CM
For CM generation, CAFs/PSCs were plated at 70–80% confluence in DMEM with 2% FBS and 1× Antibiotic/Antimycotic, 16 h prior to Veh or Ent treatment (10 μM). Culture supernatants were harvested after 48 h drug treatment, centrifuged to sediment debris (1000 g, 5 min) and applied through 0.22 μm filters. The filtered supernatants from CM were subjected to two rounds of processing with molecular weight cutoff (MWCO) centrifugation and reconstitution, to deplete the small molecule fraction (<3 kDa) containing Ent (MW 376.4). In each round, the input media were centrifuged at 3200 g for 30 min with 3 kDa MWCO filter units (Millipore); after centrifugation, the flow-throughs containing the <3 kDa fraction were discarded, and the portions enriching the >3 kDa fraction were reconstituted to the original volume with serum-free DMEM. The reconstituted fraction from the first centrifugation were subjected to a second round of processing with the same procedure. The final reconstituted CM depleted for the small-molecule fraction were analyzed for remnant Ent concentration by mass spectrometry, applied in downstream functional assays, or stored at −80 °C before use.

### CM-induced activation of STAT3 and Western blotting
The PDAC cell line p53 2.1.1 and MIA PaCa2 was seeded at 300,000 cells per well in 6-well plates and conditioned with DMEM with 2% FBS and 1× Antibiotic/Antimycotic for 16 h before serum starvation. Cells were treated with serum-free DMEM for 2 h following by 15 min CM treatment with or without an anti-LIF monoclonal antibody (4 μg/ml)[9]. Cell lysates were performed on plates with heated 1× SDS sampling buffer

(Bio-Rad, 1610147) and subjected to SDS-PAGE, followed by Western blotting using primary antibodies against pSTAT3 (Cell Signaling, 9145, 1:1000), STAT3 (Cell Signaling, 12640, 1:1000) and α-tubulin (Sigma, T6199, 1:1000), and HRP-conjugated secondary antibodies against rabbit (Santa Cruz, sc-2004, 1:5000) and mouse IgG (Santa Cruz, sc-2005, 1:5000). Chemiluminescent reaction was performed with SuperSignal West Dura substrate (Thermo, 34076). Images were acquired with a ChemiDoc XRS+ system (Bio-rad) and quantified using Image Lab software (Bio-Rad, v5.2.1).

## Spheroid formation assay
Single cell suspension from p53 2.1.1 and MIA PaCa2 cells was prepared, strained through a 70 μm filter and counted. 1000 cells were seeded in each well in ultra-low attachment 96-well plates (Corning, 3474) with 100 μl of processed CM in the presence or absence of anti-LIF antibody (4 μg/ml) or recombinant mouse LIF (0.1 μg/ml; Peprotech, 250-02). Spheroids were quantified after 8 d culture.

## Mass spectrometry analysis for Ent in CM
To extract Ent from CM samples, 50 μl of CM was diluted 1:1 with LC-MS grade water and extracted with 2.5 mL of acetonitrile/n-butylchloride (1:4, v/v) solution containing 2 pmol of d5-Apigenin as internal standard[64]. Samples were vortexed for 30 s and centrifuged at 1200 g for 10 min. The top organic layer was transferred to a glass vial and evaporated under a gentle stream of nitrogen. Samples were reconstituted in 100 μl of acetonitrile/water (50:50, v/v), vortexed for 30 s, transferred to autosampler glass vials and kept at 10 °C before injected to LC system. Targeted analysis was performed with 10 μl of injected samples on a Dionex Ultimate 3000 LC system (Thermo) coupled to a TSQ Quantiva mass spectrometer (Thermo). A Luna C18(2) C8 column (3 μm, 2 mm × 50 mm, Phenomenex) was used. Mobile phase consisting of acetonitrile/ammonium acetate (2 mM) (70:30, v/v) with 0.1 % formic acid was delivered isocratically at a flow rate of 0.2 ml/min. Mass spectrometry analyses were performed using electrospray ionization in positive ion mode, with spay voltage of 3.5 kV, ion transfer tube temperature of 325 °C, and vaporizer temperature of 275 °C. Sheath, auxiliary, and sweep gases were 35, 10 and 1, respectively. Chromatography and peak integration of the targets were verified with Skyline software (v21.1)[65]. Ent peak areas were normalized to the internal standard, and concentrations were calculated using a standard curve.

## Orthotopic co-transplantation of fibroblasts and tumor cells
Single cell suspension was prepared from mouse PDAC line p53 2.1.1 and immortalized CAF line imCAF1. Tumor cells and CAFs were mixed at a 1:5 ratio, and the mixture was further diluted with Matrigel at a 1:1 ratio (volume). $1 \times 10^4$ p53 2.1.1 and $5 \times 10^4$ control or HDAC-depleted imCAF1 cells were injected into pancreatic parenchyma in each syngeneic host (FVB/NJ). Tumor cell only injections were also performed as controls. Transplants were imaged and measured by a Vevo 3100 ultrasound imaging system (Fujifilm VisualSonics) 12 and 16 d post-transplantation, and harvested from the hosts 19 d post-transplantation. Tumor volume was calculated as 0.52 × length × width × height.

## PDAC cell proliferation curve
GFP-expressing FC1245 cells transduced with shRNAs or shEV were counted by an InFluxTM cell sorter (BD) and seeded into 96 well plates with 1000 cells per well. GFP signals representing cell numbers were measured daily by an Envision plate reader (Perkin Elmer) at 4–7 d after seeding.

## GEMM
*KP*^f/f^*C* mice (female and male) were randomized and enrolled in drug treatments around the age of 25 d. Ent was administered daily by p.o. at 5 mg/kg, and Gem at 25 mg/kg every 3 d by i.p. For tumor burden analysis, tumors were retrieved after 3-week treatments and wet tumor weight measured immediately after resection. In survival study, treatments were carried out until moribund. For scRNA-seq analysis, mice were sacrificed after Ent treatment for 15 d, and tumor immediately processed after retrieval.

## Orthotopic transplantation
To establish orthotopic transplantation model, ~100 cells from mouse PDAC line FC1245 were mixed with Matrigel (1:1 ratio by volume) and injected into pancreatic parenchyma in each syngeneic host (C57BL/6J). 10 d after transplantation, mice with successful injections were randomized and enrolled in therapeutic treatments for 2 weeks. Ent was administered daily by p.o. at 15 mg/kg, and Gem at 10 mg/kg every 3 d by i.p.

## PDX model
The PDX model was established from tumor tissues from a patient with pathologically confirmed PDAC[66]. The xenografts were minced by a sterilized scalpel, mixed in Matrigel (1:1 ratio by volume) and implanted subcutaneously into the flanks of athymic nude mice. When the tumors reached 350–450 mm³, the mice were randomized and enrolled in treatments for 3 weeks. Ent was administered by p.o. at 10 mg/kg daily. Tumors were measured twice a week using a Vernier caliper, and volume calculated (length × width² × 0.5). Mice were sacrificed when tumor volume reached 2000 mm³.

## Histology analysis
Whole pancreatic tumor specimens from the mouse models were fixed by immersing in 10% buffered formalin overnight at 4 °C and maintained in 70% ethanol for paraffin embedding. Sections (5 μm) were stained with H&E and SR according to standard protocols. Sections for immunohistochemistry (IHC) were processed with a pressure cooker for 15 min in citric acid-based antigen retrieval buffer (Vector Labs, H-3300) prior to staining. IHC staining was performed using antibodies against CK19 (Epitomic, AC-0073, 1:2000), α-SMA (Santa Cruz, sc-32251, 1:1000), Ki67 (Abcam, ab15580, 1:5000), FABP4 (Abcam, ab92501, 1:5000) and CD8a (Invitrogen, 14-0195-82, 1:100) with solutions from IHC application kits for rabbit (Cell Signaling, 13079) or mouse primary antibodies (Cell Signaling, 8125), following the manufacturer's protocols. Slides were scanned using a VS-120 virtual slide system (Olympus). Quantifications on the whole tumor sections or representative fields were performed with ImageJ (v1.54 f).

## Histopathological grading
Histopathological grading was carried out with whole tumor sections from the *KP*^f/f^*C* mice in the survival study. Tumors were graded by a pathologist with a 5-grade system, in which each grade advanced represents a 20% decrement in the percentage of gland areas from Grade 1 (very well differentiated tumors, 80–100% gland area), 2 (well differentiated, 60–80%), 3 (moderately differentiated, 40–60%), 4 (poorly differentiated, 20–40%) to 5 (undifferentiated, 0–20%).

## FACS isolation of stromal fibroblasts
Tumor retrieved from *KP*^f/f^*C* mice were minced with a razor blade and incubated with agitation for 1 h with 20 ml of freshly made digestion buffer (DMEM with 1 mg/ml collagenase IV, 1 mg/ml hyaluronidase, 0.1% soybean trypsin inhibitor, 50 U/ml DNase I and 0.125 mg/ml dispase). Dissociated tumors were filtered through 100 μm cell strainers and processed with ACK lysis buffer (Gibco). Immunofluorescence staining for FACS was performed using a standard protocol with fluorochrome-conjugated antibodies against CD45 (BV510; Biolegend, 103138, 1:200), Epcam (Alexa Fluor 647; Biolegend, 118212, 1:200) and Pdpn (APC-Cy7; Biolegend, 127418, 1:400). Sorting was performed on a

FACSAria Fusion sorter (BD). Sorted CAFs (CD45⁻ Epcam⁻ Pdpn⁺) were processed immediately for single-cell analysis or lysed in Trizol for RNA extraction.

## scRNA-seq analysis

scRNA-seq was performed with Chromium single cell kit (10x Genomics, v3), following the manufacturer's instructions. Single cell suspensions prepared by FACS were added to Chromium RT mix to achieve the loading target of 7000–10,000 cells. High-throughput sequencing was performed at a NovaSeq (Illumina) sequencer. Demultiplexed fastq files were aligned to mouse reference genome (mm10) using Cell Ranger software (10x Genomics, v4.0) with default settings. The R package Seurat (v3) was used to detect cell clusters based on gene expression and to identify subpopulation-enriched markers[67]. Cells detected with <500 and >6500 genes and those with mitochondrial genes >15% were excluded from the analysis. For data normalization, a scale factor of 10,000 was used. Canonical correlation analysis (CCA) was performed to remove batch effects. Cell clustering was performed with a 0.6 resolution and results visualized by UMAPs. Contamination cell clusters expressing well established tissue-specific markers other than those for fibroblasts were validated by bioinformatic tools (e.g., scMRMA, Enrichr) and excluded. Trajectory inference was performed with the Slingshot package[68].

## Patient cohort analysis

Stromal gene expression data from PDAC patient samples were extracted from the Gene Expression Omnibus (GEO) dataset of GSE71729 (NCBI)[17]. The integrated bulk tumor gene expression and survival data were retrieved from the PDAC database of The Cancer Genome Atlas (TCGA-PAAD)[48] with Survexpress (v2.0)[69]. Patients were stratified as high and low risk groups based on risk scores (prognostic index). For risk scores of HDAC1/2-containing complexes, the complex risk scores were represented by the average risk scores of the complex components. The inner-group $p$-values were applied to maximize the risk groups.

## Statistics

Statistical analyses were performed using Prism software (GraphPad, v10.0.2) or provided by the applied databases and algorithms. $p$-values are either stated as exact numbers ($p \geq 0.001$) or as $p < 0.001$.

## Reporting summary

Further information on research design is available in the Nature Portfolio Reporting Summary linked to this article.

# Data availability

RNA-seq, ATAC-seq and scRNA-seq data generated in this study are deposited in the Sequence Read Archive (SRA) database (NCBI) with accession number PRJNA524175. The patient dataset of stromal gene expression (GSE71729)[17] can be accessed from the GEO database of NCBI. The TCGA-PAAD dataset[48] can be accessed from the GDC data portal (https://portal.gdc.cancer.gov/projects/TCGA-PAAD). Source data are provided with this paper.

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

## Acknowledgements

We thank A. Masamune, D. Tuveson, and E. Collison for CAFs, PDAC cell lines and organoids. We thank B. Collins, T. Tseng, H. Juguilon, P. Medina, E. Tilley for technical assistance, and L. Ong and C. Brondos for administrative assistance. This work was funded by grants from the Lustgarten Foundation (Distinguished Scholar Award to R.M.E and 122215393-02), the Don and Lorraine Freeberg Foundation, the David C Copley Foundation, the Wasily Family Foundation, a Stand Up To Cancer-Cancer Research UK-Lustgarten Foundation Pancreatic Cancer Dream Team Research Grant (SU2C-AACR-DT-20-16), the NOMIS

Foundation (Science of Health) and NIH grants (CA220468 and CA265762). This work was supported by the Next Generation Sequencing, Flow Cytometry, and Advanced Biophotonics Cores in the Salk Institute with funding from NIH-NCI (CCSG: P30 014195) and the Waitt Foundation. G.L. was supported by a Ruth L. Kirschstein National Research Service Award (F32CA217033). M.L.T. was supported by a Life Sciences Research Foundation Fellowship. C.E.A. was a Robert Black Fellow of the Damon Runyon Cancer Research Foundation (DRG-2244-16) and was supported by an Institutional Research Training Grant (5T32CA009370). T.H. is a Frank and Else Schilling American Cancer Society Professor and the Renato Dulbecco Chair in Cancer Research, and work in his group was supported by an NIH grant (CA082683), a Lustgarten Foundation Award (552873), and a William Isacoff Research Foundation Award. R.M.E. holds the March of Dimes Chair in Molecular and Developmental Biology at the Salk Institute.

## Author contributions

G.L., M.D., and R.M.E. conceived, designed and supervised the study. G.L. performed or participated in all the experiments and analyses with the technical assistance from D.C.N., G.E., S.B., and E.B. T.G.O. analyzed the scRNA-seq and patient expression data. N.H. analyzed the ATAC-seq data. Y.S. participated in immunoassay. Y.S. and T.H. provided the KPf/fC mice and α-LIF antibody. C.F. performed histopathological analysis. S.N. performed PDX study. A.F.M.P. performed mass spectrometry analysis. Y.D. participated in RNA-seq and scRNA-seq experiments. C.L. analyzed the RNA-seq data. R.T.Y. coordinated data curation. H.T., M.L.T., C.E.A., and Y.L. provided scientific input and experimental advice. T.H., D.D.E., H.H., and D.D.V.H. provided scientific guidance. G.L., R.T.Y., A.R.A., M.D., and R.M.E. prepared the manuscript.

## Competing interests

R.M.E. and M.D. are co-founders of a company developing entinostat. The remaining authors declare no competing interests.
