## [Peer Review File · Nature Communications]

Inhibiting stromal Class I HDACS curbs pancreatic cancer progressionEditorial Note: This manuscript has been previously reviewed at another journal that is not operating a transparent peer review scheme. This document only contains reviewer comments and rebuttal letters for versions considered at *Nature Communications*.

REVIEWER COMMENTS

Reviewer #1 (Remarks to the Author):

The authors have adequately addressed to all my comments; the revised manuscript has made substantial improvements upon the previous submission- this work will make a great addition to pancreatic research field.

1 minor correction: the authors referred Fig. 5i & j (page 14) which do not exist in the main Figure. To my understanding, this referral should be Extended Data Fig. 7b & c?

Reviewer #2 (Remarks to the Author):

In their revised manuscript the authors nicely addressed most of my previous comments.

Two minor comments to consider:

1. In response to major comment 4, the authors added a functional assay in which Sirius red staining was performed on whole tumor sections from KPf/fC mice under Veh or Ent treatment. The authors indicate the tumors under Ent treatment showed a reduction in the volume of CK19+ and a-SMA+ cells, and reduces KI67+ proliferative cells. Sirius red staining may be reduced by fewer cells and smaller tumor volumes, and it does not necessarily indicate less myofibroblast activity.
2. The title of the revised manuscript is somewhat general and the word “resetting” is too strong.

Reviewer #3 (Remarks to the Author):

The revised manuscript by Liang et al. is substantially improved and the authors should be commended for performing a large number of additional experiments and analyses. The major concerns previously raised have been adequately addressed; in particular, data showing consistent behavior of the different models used has strengthened the conclusions, and the experiments involving HDACi-depleted medium in vitro and HDACs-depleted cells in vivo are now convincing. This is now a well-executed study that can be of wide interest to the community, especially in the PDAC field.

Due to the characterized time points, I still think that the initial part mainly captures the consequences of cell proliferation driven by culture conditions, rather than the mechanisms driving PSC activation. Nevertheless, the fact that at least some of those changes can be prevented by HDACi treatment, establishes the basis for the subsequent experiments.

This referee is an expert of epigenetic regulation of gene expression, and knows what ATAC-seq detects and what GSEA q-value are, as well as what biological conclusions can be reliably drawn based on those. Long responses to very specific requests do not strengthen weak data. Since it is not key to the main message of the paper, I will not highlight all the concerns I have about the RNA-seq and ATAC-seq analysis, but some additional plots need to be provided before the paper can be accepted for publication.

1 (Related to previous point 1). Please provide a UMAP similar to those shown in Ext data 8 d,e visualizing the single cell expression levels for SRF, FOXM1, and HDAC1,2,3 in vehicle and HDACi treated samples across the different subpopulations.

2 (Related to previous point 2). Please show in fig. 2e and f heatmaps and metagene profiles for each technical replicate to be able to assess reproducibility of global trends.

Furthermore, show similar plots for: a) gene sets corresponding to myofibroblast, proliferation and lipid metabolism transcriptional program, centered on TSS, providing a table with the genes belonging to each program; b) for up- and down-regulated used for Fig. 2g.

If these plots do not fully support current conclusions, text should be thoroughly edited to avoid overstatements.

3 (Related to previous point 5). The confusion about the analysis of patient data is partly due to term “normal”, which I assumed meant normal PSCs. The edited text is now clearer but I suggest using normal-stroma PDAC subtype.

Reviewer #1 (Remarks to the Author)

The authors have adequately addressed to all my comments; the revised manuscript has made substantial improvements upon the previous submission- this work will make a great addition to pancreatic research field.

We appreciate the support of the Reviewer.

1 minor correction: the authors referred Fig. 5i & j (page 14) which do not exist in the main Figure. To my understanding, this referral should be Extended Data Fig. 7b & c?

Thanks for the correction. The referral indeed should be Extended Data Fig. 7b and c (now Supplementary Fig. 8b and c).

Reviewer #2 (Remarks to the Author)

In their revised manuscript the authors nicely addressed most of my previous comments.

We thank the Reviewer for recognizing our efforts in addressing the comments.

Two minor comments to consider:

1. In response to major comment 4, the authors added a functional assay in which Sirius red staining was performed on whole tumor sections from KPf/fC mice under Veh or Ent treatment. The authors indicate the tumors under Ent treatment showed a reduction in the volume of CK19+ and a-SMA+ cells, and reduces KI67+ proliferative cells. Sirius red staining may be reduced by fewer cells and smaller tumor volumes, and it does not necessarily indicate less myofibroblast activity.

Thank you for the comment. We agree that lower collagen content seen under Ent treatment might be due to reduced tumor content, fewer activated myofibroblasts, lower fibrogenic activity in myofibroblasts, or more likely, a combination of these factors under Ent treatment.

2. The title of the revised manuscript is somewhat general and the word “resetting” is too strong.

In light of the Reviewer’s comment, the title is changed to “Inhibiting stromal Class I HDACs curbs pancreatic cancer progression”.

Reviewer #3 (Remarks to the Author)

The revised manuscript by Liang et al. is substantially improved and the authors should be commended for performing a large number of additional experiments and analyses. The major concerns previously raised have been adequately addressed; in particular, data showing consistent behavior of the different models used has strengthened the conclusions, and the experiments involving HDACi-depleted medium in vitro and HDACs-depleted cells in vivo are now convincing. This is now a well-executed study that can be of wide interest to the community, especially in the PDAC field.

Due to the characterized time points, I still think that the initial part mainly captures the consequences of cell proliferation driven by culture conditions, rather than the mechanisms driving PSC activation. Nevertheless, the fact that at least some of those changes can be prevented by HDACi treatment, establishes the basis for the subsequent experiments.

We thank the Reviewer for acknowledging our efforts and offering support for the revised manuscript.

This referee is an expert of epigenetic regulation of gene expression, and knows what ATAC-seq detects and what GSEA q-value are, as well as what biological conclusions can be reliably drawn based on those. Long responses to very specific requests do not strengthen weak data. Since it is not key to the main message of the paper, I will not highlight all the

concerns I have about the RNA-seq and ATAC-seq analysis, but some additional plots need to be provided before the paper can be accepted for publication.

We clearly acknowledge your expertise and apologize for the long responses.

1 (Related to previous point 1). Please provide a UMAP similar to those shown in Ext data 8 d,e visualizing the single cell expression levels for SRF, FOXM1, and HDAC1,2,3 in vehicle and HDACi treated samples across the different subpopulations.

As requested, we now provide UMAPs showing *Srf*, *Foxm1*, and *Hdac1*, 2 and 3 expression (Response Fig. 1a) and a complementary heatmap of normalized average expression values that reveal CAF subpopulation-specific reductions with Ent-treatment (Response Fig. 1b). Response Fig. 1b is incorporated into Supplementary Fig. 9j (*Srf*), m (*Foxm1*) and n (*Hdac1*, 2, 3) in the revised manuscript.

Response Fig. 1 | a, UMAPs showing the expression of selected TF and HDAC genes in CAFs from Veh or Ent-treated *KP^{fl/c}* mice. **b**, Heatmap showing the relative expression of selected TF and HDAC genes in total CAFs (pool) and CAF subpopulations from Veh or Ent-treated *KP^{fl/c}* mice.

2 (Related to previous point 2). Please show in fig. 2e and f heatmaps and metagene profiles for each technical replicate to be able to assess reproducibility of global trends.

As requested, we now provide heatmaps and metagene profiles for the replicates used to generate the original Fig. 2e and f, demonstrating the reproducibility of the ATAC-seq data (Response Fig. 2a and b). Response Fig. 2a and b are now shown in Fig. 2e and f.

Furthermore, show similar plots for: a) gene sets corresponding to myofibroblast, proliferation and lipid metabolism transcriptional program, centered on TSS, providing a table with the genes belonging to each program; b) for up- and down-regulated used for Fig. 2g. *Note: Judging from the figures and the context of the comment, we assume the Reviewer might have intended to refer to Fig. 1g, instead of Fig. 2g.*

Response Fig. 2 | a, b, Heatmap (a) and histogram (b) showing individual replicates at the genomic sites with accessibility upregulated in activated PSCs. **c-f**, Heatmaps and Histograms showing the genomic accessibility patterns associated with selected gene sets, including genes related to myofibroblast identity (c) proliferation (d), or lipid metabolism (e), as well as genes upregulated in PSC activation and downregulated by Ent (e).

Changes in genomic accessibility with PSC activation and upon Ent treatment at myofibroblast, proliferation, and lipid metabolism genes are respectively provided in Response Figs. 2c, d, and e; while those at genes upregulated by PSC activation and downregulated by Ent are shown in Response Fig. 2f (the gene set from Fig. 1g). The data from myofibroblast and proliferation genes (Response Fig. 2c, d), as well as a broader set of genes upregulated by PSC activation and downregulated by Ent (Response Fig. 2f), recapitulate the patterns already exemplified by representative myofibroblast and proliferation genes in our manuscript (Fig. 2j), further supporting our conclusion that genomic accessibility correlates with expression changes at genes driving PSC activation. At lipid metabolism genes, however, no correlation is evident between changes in genomic accessibility (increased post-activation but insensitive to Ent) and transcriptional output (downregulated post-activation and upregulated by Ent) (Response Fig. 2e). This result, plus the data from individual representative genes (Supplementary Fig. 4h), supports our conclusion that no consistent correlation between expression and genomic accessibility is observed for this gene set. These analyses have now been incorporated into the revised manuscript (Response Fig. 2c, d in Supplementary Fig. 4b, c; Response Fig. 2e in Supplementary Fig. 4g; Response Fig. 2f in Supplementary Fig. 4a), along with the requested gene sets (Supplementary Data 5).

If these plots do not fully support current conclusions, text should be thoroughly edited to avoid overstatements.

3 (Related to previous point 5). The confusion about the analysis of patient data is partly due to term “normal”, which I assumed meant normal PSCs. The edited text is now clearer but I suggest using normal-stroma PDAC subtype.

We have now clarified that the patient cohort is stratified by normal- or activated-stroma subtype in the text and figure legend (Supplementary Fig. 10a).

REVIEWERS' COMMENTS

Reviewer #3 (Remarks to the Author):

The authors have addressed the remaining points and I recommend publication.